# Restricted Global-Aware Graph Filters Bridging GNNs and Transformer for Node Classification

**Jingyuan Zhang**[1,2]    **Xin Wang**[1,2]    **Lei Yu**[1,2]    **Zhirong Huang**[1,2]
**Li Yang**[1,*]    **Fengjun Zhang**[1,*]

[1] Institute of Software, Chinese Academy of Sciences
[2] University of Chinese Academy of Sciences
{zhangjingyuan2023, wangxin, yulei2022, huangzhirong2022}@iscas.ac.cn
{yangli2017, fengjun}@iscas.ac.cn

## Abstract

Transformers have been widely regarded as a promising direction for breaking through the performance bottlenecks of Graph Neural Networks (GNNs), primarily due to their global receptive fields. However, a recent empirical study suggests that tuned classical GNNs can match or even outperform state-of-the-art Graph Transformers (GTs) on standard node classification benchmarks. Motivated by this fact, we deconstruct several representative GTs to examine how global attention components influence node representations. We find that the global attention module does not provide significant performance gains and may even exacerbate test error oscillations. Consequently, we consider that the Transformer is barely able to learn connectivity patterns that meaningfully complement the original graph topology. Interestingly, we further observe that mitigating such oscillations enables the Transformer to improve generalization in GNNs. In a nutshell, we reinterpret the Transformer through the lens of graph spectrum and reformulate it as a global-aware graph filter with band-pass characteristics and linear complexity. This unique perspective introduces multi-channel filtering constraints that effectively suppress test error oscillations. Extensive experiments (17 homophilous, heterophilous graphs) provide comprehensive empirical evidence for our perspective. This work clarifies the role of Transformers in GNNs and suggests that advancing modern GNN research may still require a return to the graph itself.

## 1  Introduction

Graph Neural Networks (GNNs) [20, 26, 57, 8, 41, 43, 64, 67] have demonstrated powerful modeling capabilities on graph-structured data and have been widely applied to various tasks [16, 35, 33, 68, 60, 44] such as node classification. Most existing GNNs follow a message-passing paradigm [18], where messages are exchanged between a target node and its neighbors to integrate node features with graph topology, exhibiting strong spatial locality. Naturally, this local message exchange has raised concerns about the lack of global awareness, which is often viewed as a performance bottleneck for GNNs. To address this limitation, some studies [42, 40] have explored incorporating Transformers into graph learning, giving rise to Graph Transformers (GTs). However, it is unclear to what extent Transformers enhance GNNs.

In the literature, research on GTs has evolved through three distinct stages: **(i) Expressive Power.** In the early stage, GTs primarily leveraged the attention mechanism in Transformers [56] to establish global receptive fields, alleviating the limitations on expressive power caused by over-smoothing [30, 5] and over-squashing [1, 14]. To enhance model expressiveness, various forms of positional encoding

---

*Corresponding Author

(PE, including Laplacian spectral encoding [4, 28], learnable spectral encoding [15], and both global and local positional encoding [47].) or structural encoding (SE, including node degree centrality [66], node subgraph representations [6], pairwise distance and path encoding [70], and random walk-based encoding [37].) were introduced. **(ii) Scalability.** GTs have shown promising encoding capabilities in comparison to GNNs. However, the computational complexity of GNNs typically scales linearly with the number of edges ($\mathcal{O}(|\mathcal{E}|)$), whereas GTs generally incur a quadratic complexity ($\mathcal{O}(|\mathcal{V}|^2)$) due to the computation of attention score matrices. This prompted the development of linear or sparse GTs [10, 69, 52, 27, 62, 61]. Recent advances have shown that state-of-the-art GTs can strike a balance between expressivity and efficiency. For example, [63] achieves competitive performance with a single-layer global attention network of linear complexity. [13] combines high-order polynomial expressiveness with linear computational complexity. [53] adopts a two-stage process to further sparsify the attention mechanism, thereby reducing memory overhead. **(iii) Effectiveness (Actual Gains).** At this stage, GTs appear to be relatively mature. However, a recent empirical study [36] cast doubt on their actual benefits, showing that fine-tuned classical GNNs can perform on par with or even surpass the SOTA GTs. This finding challenges the fundamental premise of GTs research: *Does the Transformer truly enhance the performance of GNNs?* As there is currently no definitive answer, the field faces significant challenges that warrant deeper reflection.

In this work, *we argue that Transformers may not empirically boost the performance upper bound of GNNs, but they can serve as regularization components to enhance generalization of GNNs.* This conclusion stems from a unique perspective on understanding Transformers. Prior efforts largely inherit the standard Transformer framework (i.e., the $\mathbf{Q}\mathbf{K}^T\mathbf{V}$ paradigm), employing attention either as a supplement to graph topology or as a downstream encoder following GNNs. However, based on our experimental observations (as shown in Figure 1), the former often introduces substantial topological noise, where redundant global signals contribute little to optimization or generalization. As for the latter, building upon the upstream GNN results, the Transformer merely re-encodes already-captured information, limiting its effectiveness. From the lens of GNNs, Transformer can be seen as learning a new shift operator over a complete graph (i.e., the attention matrix) and applying it for low-pass filtering (i.e., neighbor aggregation). If key inductive biases lie in localized structures, this relaxed low-pass channel setting may lead to a form of representational "laziness" in the attention mechanism. Enforcing sparsity in attention to contain such locality is theoretically appealing, but computing the full $\mathbf{Q}\mathbf{K}^T$ remains costly. Therefore, our focus naturally shifts to diversifying the filtering channels, offering a new path for bridging GNNs and Transformers.

Based on the above analysis, we innovatively extend the Transformer into a band-pass global-aware graph filter — G$^2$Former, departing from the conventional cascade fashion. The band-pass property allows the Transformer to adaptively learn graph spectrum determined by the graph topology under different channels. This channel diversity encourages the attention mechanism to move beyond pure similarity modeling between node pairs, thus exerting a certain regularization to facilitate better generalization for GNNs. The details are as follows:

- Rather than advocating a naive combination of GNNs and Transformers, we propose G$^2$Former, a structurally extended architecture from the perspective of graph signal processing. Under multi-channels, G$^2$Former induces Transformers to generate guided noise that enhances node features, thereby benefiting the inference of downstream GNNs. This constitutes a novel **global-to-local** framework that effectively addresses the aforementioned "laziness" issue.

- G$^2$Former theoretically supports band-pass filtering at arbitrary frequency bands, thereby satisfying constraint requirements and adapting to diverse graph structures. Attention mechanism enables a node feature–driven and learnable graph spectrum, challenging the traditional view of topology as a fixed prior in graph filtering. We design and discuss two channel initialization strategies to highlight the importance of channel constraints. And G$^2$Former retains linear computational complexity.

- Extensive experiments on 17 datasets of real-world node classification - covering homophilous, heterophilous, and large-scale graphs with millions of nodes - demonstrate our claims. We hope this work helps break the current dilemma in GTs research and revisit the positioning of Transformers for the field. Our code is available at `https://github.com/Thankstaro/G-2Former`.

## 2 Background

This section presents preliminaries in two relevant areas: Transformers and graph filtering.

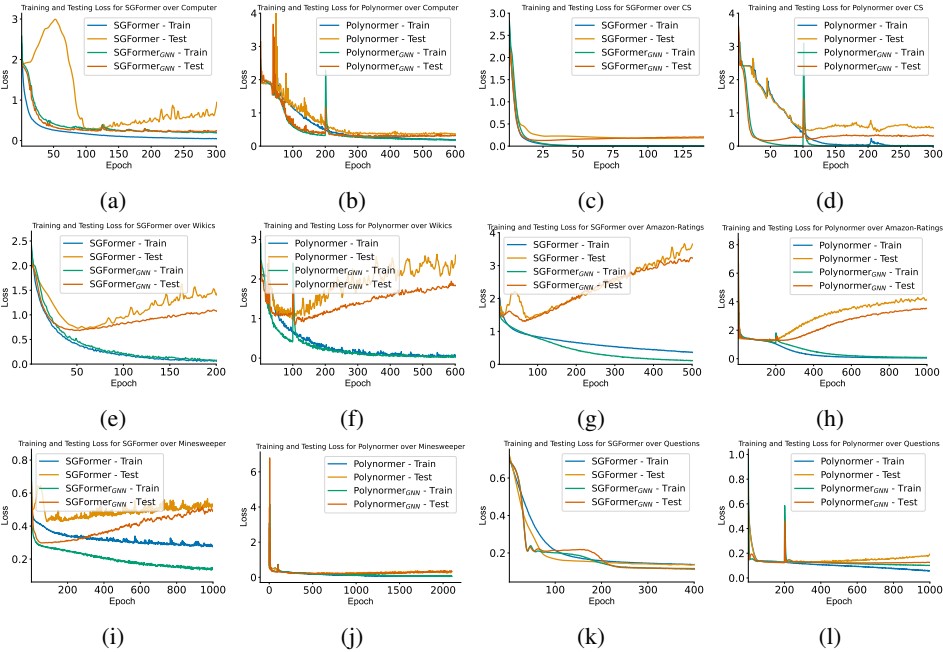

Figure 1: SGFormer [63] and Polynormer [13] correspond to two distinct GT architectures: local-and-global and local-to-global, respectively. We report their training and test cross-entropy losses across various datasets. SGFormer$_{-GNN}$ retains only the fine-tuned GCN backbone, while Polynormer$_{-GNN}$ keeps only its local attention mechanism (which is GNN-like). (a)-(f) correspond to homophilous graphs, and (g)-(l) to heterophilous graphs. These two GTs fail to provide significant test improvements for their GNNs and even underperform compared to standalone GNNs on some datasets.

**Transformers.** For a general self-attention Transformer architecture [56], each layer computes attention scores between every pair of samples, yielding an attention coefficient matrix. This matrix is then used to linearly reconstruct the sample features and update the representation at the current layer. For example, given a sample feature matrix $\mathbf{X} \in \mathbb{R}^{n \times d}$, the self-attention can be computed in the following two ways:

$$\mathbf{X}' = \text{softmax}(\frac{\mathbf{Q}\mathbf{K}^T}{\sqrt{d}})\mathbf{V}, \mathbf{X}' = \frac{\sigma(\mathbf{Q})(\sigma(\mathbf{K})^T\mathbf{V})}{\sigma(\mathbf{Q})(\sigma(\mathbf{K})^T\mathbf{1})}, \tag{1}$$

where $\mathbf{Q} = \mathbf{X}\mathbf{W}_Q, \mathbf{K} = \mathbf{X}\mathbf{W}_K, \mathbf{V} = \mathbf{X}\mathbf{W}_V$ are the query, key and value matrices. $\mathbf{W}_Q \in \mathbb{R}^{d \times d'}, \mathbf{W}_K \in \mathbb{R}^{d \times d'}, \mathbf{W}_V \in \mathbb{R}^{d \times d'}$ are trainable matrices, $\mathbf{1} \in \mathbb{R}^{n \times 1}$ is an all-one vector, and $\sigma$ is the sigmoid function. The first equation is the commonly used explicit computation of the self-attention coefficient matrix, but it is computationally expensive, requiring $\mathcal{O}(n^2)$ operations. The second equation is a linear attention [25] formulation, which leverages a simple kernel trick to reorder the computation and reduce the complexity to $\mathcal{O}(n)$. Division by $\sigma(\mathbf{Q})(\sigma(\mathbf{K})^T\mathbf{1})$ ensures that each row of the attention matrix sums to 1. GTs typically integrate the graph adjacency matrix with the attention matrix, or apply graph convolution before feeding the features into the Transformer.

**Graph Filtering.** For an undirected attributed graph $\mathcal{G} = (\mathcal{E}, \mathcal{V}, \mathbf{X})$, $\mathcal{E}, \mathcal{V}$ are the sets of edges and nodes, $\mathbf{X}$ is the node attribute matrix. Let $\mathbf{A}$ be the adjacency matrix of $\mathcal{G}$, the symmetric normalized Laplacian $\tilde{\mathbf{L}} = \mathbf{D}^{-1/2}(\mathbf{D} - \mathbf{A})\mathbf{D}^{-1/2}$, $\mathbf{D}$ is the degree matrix of $\mathbf{A}$. Perform the eigenvalue decomposition on $\tilde{\mathbf{L}}$, i.e. $\tilde{\mathbf{L}} = \mathbf{U}\mathbf{\Lambda}\mathbf{U}^T$. $\mathbf{U}$ is the eigenvector matrix, $\mathbf{\Lambda} = \text{diag}([\lambda_1, \lambda_2, ..., \lambda_n])$. $\lambda_i$ is $i$-th eigenvalue of $\tilde{\mathbf{L}}$, $0 = \lambda_1 \leq \lambda_2 \cdots \leq \lambda_n \leq 2$. Assume that $\mathbf{x} \in \mathbb{R}^{n \times 1}$ is a signal of $\mathcal{G}$, $g(\mathbf{\Lambda})$ is the frequency response function, the graph filtering can be formulated as:

$$\mathcal{F}(\tilde{\mathbf{L}})\mathbf{x} = \mathbf{U}g(\mathbf{\Lambda})\mathbf{U}^T\mathbf{x}. \tag{2}$$

First, transform $\mathbf{x}$ into the spectral domain via the graph Fourier transform [49] $\hat{\mathbf{x}} = \mathbf{U}^T\mathbf{x}$. $\lambda_i$ indicates the frequency magnitude of the graph Fourier basis $\mathbf{U}_i$, a designed $g(\mathbf{\Lambda})$ is applied to amplify or suppress corresponding frequency components, and the signal is then mapped back to

the spatial domain via the inverse graph Fourier transform $\mathbf{U}\hat{x}$ to complete the filtering process. $\mathcal{F}(\tilde{\mathbf{L}})$ is referred to as the graph filter [11, 45]. Combining multiple distinct designed $\mathcal{F}(\tilde{\mathbf{L}})$ enables multi-channel [54] or mixed-channel [34] filtering.

# 3 Methodology

In this section, we provide a detailed explanation of the proposed method. In Section 3.1, we describe how Transformer is extended into a global-aware graph filter with band-pass characteristics, along with two different strategies for channel initialization. In Section 3.2, we present the method for generating guided noisy samples from multiple channels and performing graph-guided filtering to enhance node features for downstream GNN training or inference. Finally, we introduce the $G^2$Former framework, its optimization objective, and provide theoretical analysis (including the computational complexity) in Section 3.3 and 3.4.

## 3.1 Global-Aware Graph Filters

As shown in Equation 1, self-attention can be viewed as a form of topology reconstruction (graph structure learning) followed by message passing over the updated topology. From the perspective of GNNs, the behaviors of both are highly similar. To address the "laziness" that arises when combining it with GNNs and to fully leverage their global receptive field, we develop the band-pass property of the Transformer. We follow the common setting of Section 2 that denote $\mathbf{Q}'_i = \frac{\sigma(\mathbf{Q})_i}{\|\sigma(\mathbf{Q})_i\|_2}, \mathbf{K}'_i = \frac{\sigma(\mathbf{K})_i}{\|\sigma(\mathbf{K})_i\|_2}, \mathbf{Q}' \in \mathbb{R}^{n \times d'}_{\geq 0}, \mathbf{K}' \in \mathbb{R}^{n \times d'}_{\geq 0}$, $\sigma$ is the sigmoid function. Let $\mathbf{A}_S = \mathbf{Q}'\mathbf{K}'^T$, if $\mathbf{Q}'$ and $\mathbf{K}'$ do not share parameters, $\mathbf{A}_S = \frac{1}{2}(\mathbf{Q}'\mathbf{K}'^T + \mathbf{K}'\mathbf{Q}'^T)$, ensuring that $\mathbf{A}_S$ is a symmetric matrix. Similarly, $\tilde{\mathbf{L}}_S = \mathbf{D}_S^{-1/2}(\mathbf{D}_S - \mathbf{A}_S)\mathbf{D}_S^{-1/2}$, $\mathbf{D}_S$ is the degree matrix of $\mathbf{A}_S$. We define a band-pass global-aware graph filter as follows:

$$\mathcal{F}(\tilde{\mathbf{L}}_S; \alpha, \beta, \Theta) = \frac{(\tilde{\mathbf{L}}_S)^\alpha (2\mathbf{I} - \tilde{\mathbf{L}}_S)^\beta}{\int_0^2 g_S(\lambda)\mathrm{d}\lambda} = \frac{(\tilde{\mathbf{L}}_S)^\alpha (2\mathbf{I} - \tilde{\mathbf{L}}_S)^\beta}{\int_0^2 \lambda^\alpha (2 - \lambda)^\beta \mathrm{d}\lambda}, \tag{3}$$

where $\Theta = \{\mathbf{W}_Q, \mathbf{W}_K\}$, $\mathbf{I}$ is the identity matrix, $g_S(\lambda)$ is the frequency response function of $(\tilde{\mathbf{L}}_S)^\alpha (2\mathbf{I} - \tilde{\mathbf{L}}_S)^\beta$. $\alpha, \beta \in \mathbb{N}$ are the band control coefficients. $\Theta$ enables the filter to adaptively learn the graph topology, thereby adjusting the graph spectrum. $\int_0^2 g_S(\lambda)\mathrm{d}\lambda$ represents the area under the curve of $g_S(\lambda)$, acting as a normalization term for $\mathcal{F}(\tilde{\mathbf{L}}_S; \alpha, \beta, \Theta)$ to prevent numerical explosion due to over-amplification, it can be written as:

$$\int_0^2 \lambda^\alpha (2 - \lambda)^\beta \mathrm{d}\lambda = 2^\alpha \cdot 2^\beta \int_0^2 (\frac{\lambda}{2})^\alpha (\frac{2 - \lambda}{2})^\beta \mathrm{d}\lambda. \tag{4}$$

Let $\lambda/2 = t$, it can be rewritten as:

$$2^\alpha \cdot 2^\beta \int_0^2 (\frac{\lambda}{2})^\alpha (\frac{2 - \lambda}{2})^\beta \mathrm{d}\lambda = 2^{\alpha+\beta+1} \int_0^1 t^\alpha (1 - t)^\beta \mathrm{d}t. \tag{5}$$

Note that $\int_0^1 t^\alpha (1 - t)^\beta \mathrm{d}t$ is the Beta function $B(\alpha + 1, \beta + 1)$. According to the relationship [12] between the Beta function and the Gamma function $\Gamma$, we can get:

$$2^{\alpha+\beta+1} \int_0^1 t^\alpha (1 - t)^\beta \mathrm{d}t = 2^{\alpha+\beta+1} \cdot \frac{\Gamma(\alpha + 1)\Gamma(\beta + 1)}{\Gamma(\alpha + \beta + 2)}. \tag{6}$$

Hence, the final expression of the filter is:

$$\mathcal{F}(\tilde{\mathbf{L}}_S; \alpha, \beta, \Theta) = \frac{\Gamma(\alpha + \beta + 2)(\tilde{\mathbf{L}}_S)^\alpha (2\mathbf{I} - \tilde{\mathbf{L}}_S)^\beta}{\Gamma(\alpha + 1)\Gamma(\beta + 1)2^{\alpha+\beta+1}}, \tag{7}$$

where $\Gamma(\alpha + 1) = \alpha!$.

As is well known, $\tilde{\mathbf{L}}_S, 2\mathbf{I} - \tilde{\mathbf{L}}_S$ correspond to high-pass and low-pass channels, respectively. Therefore, when $\alpha < \beta$, $\mathcal{F}(\tilde{\mathbf{L}}_S; \alpha, \beta, \Theta)$ focus on a certain low-frequency range; conversely, it passes signals

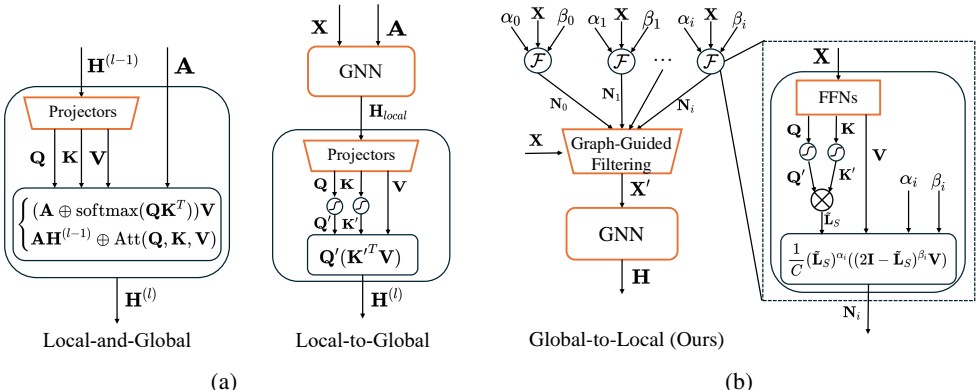

Figure 2: The architecture of $G^2$Former and its comparison with previous works. The modules marked in orange contain learnable parameters. $\oplus$ denotes the fusion of topological information with the self-attention mechanism. Att: self-attention computation. $\otimes$ represents the computation of $\tilde{\mathbf{L}}_S$ as described in Section 3.1, and $C$ corresponds to Equation 6.

in a certain high-frequency band. We can adjust the frequency band that the filter emphasizes by tuning $\alpha, \beta$. For $\mathcal{F}(\tilde{\mathbf{L}}_S; \alpha > 0, 0, \Theta)$ and $\mathcal{F}(\tilde{\mathbf{L}}_S; 0, \beta > 0, \Theta)$, they are also high-pass and low-pass channels, respectively. In our implementation, two feedforward networks (FFNs), $f_{\theta_1}$ and $f_{\theta_2}$, are initialized. Specifically, $f_{\theta_1}$ serves as a replacement for $\{\mathbf{W}_Q, \mathbf{W}_K\}$ ($\mathbf{Q} = \mathbf{K} = f_{\theta_1}(\mathbf{X})$), while $f_{\theta_2}$ maps $\mathbf{X}$ into a latent space prior to filtering, and $\Theta = \{\theta_1, \theta_2\}$ now.

**Channel Initialization.** We introduce two channel initialization strategies: full-spectrum and low-spectrum (in Figure 3). We use $\hat{\lambda}$ to denote the pass-band of a specific channel, and $g(\hat{\lambda})$ represents the average frequency response within that channel. Assuming $\lambda_{thre} = 1$ as the threshold, $[0, 1]$ corresponds to the low-frequency band, while $(1, 2]$ represents the high-frequency band. The full-spectrum filter bank is denoted as $\{\mathcal{F}(\tilde{\mathbf{L}}_S; i, \alpha + \beta - i, \Theta) \mid i \in \{0, .., \alpha + \beta\}\}$, and low-spectrum filter bank is $\{\mathcal{F}(\tilde{\mathbf{L}}_S; i, \alpha + \beta, \Theta) \mid i \in \{0, .., \alpha + \beta\}\}$. In general, node classification tasks primarily rely on low-frequency components, which may make the low-spectrum initialization sufficient without the need to incorporate high-frequency signals. However, such a compromise can be detrimental to attention mechanisms. Focusing solely on low-frequency signals may still lead to underconstrained behavior, whereas full-spectrum coverage provides a more principled solution. A detailed discussion can be found in Section 4.3.

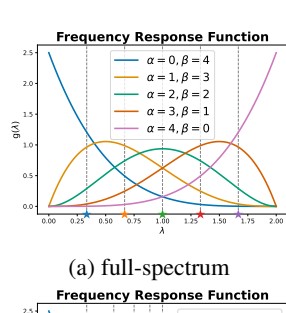

(a) full-spectrum

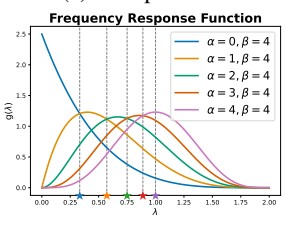

(b) low-spectrum

Figure 3: Frequency response functions of the two channel initialization strategies (with $\max(\alpha, \beta) = 4$ as an example). The pass-band $\hat{\lambda}$ of each channel is indicated by dashed lines and marked with penta-grams in corresponding colors.

### 3.2 Graph-Guided Filtering

Inspired by guided filtering [21], we attempt to use the graph filter introduced in Section 3.1 to generate guided noisy samples that enhance the original node features and facilitate downstream GNNs training and inference. By adjusting $\alpha, \beta$, the guided noise encompasses features from various channels (low-pass, high-pass, or band-pass) and retains a global perspective. This design aims to circumvent model overfitting caused by a single relaxation channel. In the field of image processing, the guided filter is defined as follows:

$$q_i = a_k I_i + b_k, \forall i \in w_k. \tag{8}$$

This means that the filtered image $q$ is a linear transformation of the guidance (or guided noise) $I$ within a local window $w_k$. With reference to the above equation, we define graph-guided filtering kernel as follows:

$$\mathbf{X}'_i = a_i \mathbf{N}_i + (1 - a_i)\mathbf{X}_i, a_i \in [0, 1], i = 0, 1, .., n - 1, \tag{9}$$

where $a_i$ is the weight. In this case, the filtered attribute $\mathbf{X}'$ is the linearly weighted sum of the original attribute $\mathbf{X}$ and the guided noise $\mathbf{N}$. Graph-guided filtering is not determined solely by the guided noise; the incorporation of original features serves as a residual connection, which facilitates more effective model training. When $\alpha, \beta$ are fixed, the guided noise under different channel initialization strategies can be computed as follows:

$$
\mathbf{N} = \begin{cases} \sum_{i=0}^{\alpha+\beta} \mathcal{F}(\tilde{\mathbf{L}}_S; i, \alpha + \beta - i, \Theta) f_{\theta_2}(\mathbf{X}) \\ \sum_{i=0}^{\alpha+\beta} \mathcal{F}(\tilde{\mathbf{L}}_S; i, \alpha + \beta, \Theta) f_{\theta_2}(\mathbf{X}) \end{cases}. \tag{10}
$$

The weight $a \in \mathbb{R}^{n \times 1}$ in Equation 9 is provided by the model and is updated at each iteration. To obtain the wights, we initialize a predictor $f_{\theta_g}$, parameterized as a fully connected layer, $a_i = \max(f_{\theta_g}(\mathbf{N}_i))$. $f_{\theta_g}$ is optimized by computing the loss with respect to the training labels, which differs from the way it is derived in Equation 8. While traditional guided filtering minimizes the error between the filtered and original images, our graph-guided filtering adaptively assigns weights based on the confidence of the guided noise. This filtering approach injects the guidance into $\mathbf{X}$, balancing the correlation of the filtered result with both the original features and the guidance.

### 3.3 The G$^2$Former Architecture

This section presents the architecture followed by G$^2$Former and highlights its differences from previous works. Subsequently, we introduce the optimization objective of the model.

**Global-to-Local Scheme.** Previous GT architectures can be mainly categorized into two classes: Local-and-Global and Local-to-Global. As analyzed in Section 1, we propose a Global-to-Local scheme. The "Global" component is placed upfront to generate guidance via the global graph filter bank, enabling graph-guided filtering. This is then followed by downstream "Local" encoding. Our design is orthogonal to the choice of downstream GNNs, allowing users to flexibly plug in various customized GNNs. At this stage, the Transformer generates guidance under the constraints of multiple channels, which distinguishes it from the conventional stacked or cascade encoding approach with GNNs. This design effectively improves the coordination between the two components. The detailed architecture is illustrated in Figure 2.

**Optimization Objective.** In general, the optimization objective of typical GTs only involves the cross-entropy loss between the predicted probabilities and the ground-truth labels. In contrast, G$^2$Former introduces an additional cross-entropy loss term. As shown in Equation 9, obtaining the weight $a$ for generating the guidance also requires supervision from the labels. We initialize a separate predictor $f_{\theta_p}, f_{\theta_g}$ (as mentioned in Section 3.2) for the final encoded representation $\text{GNN}(\mathbf{X}')$ and the guided noise $\mathbf{N}$, respectively, to compute their corresponding loss terms. Let $P = f_{\theta_p}(\text{GNN}(\mathbf{X}')), P' = f_{\theta_g}(\mathbf{N})$ the overall loss can be expressed as:

$$
\mathcal{L} = - \sum_{v \in \mathcal{V}_{train}} \sum_{k=0}^{K-1} Y_v[k] \ln P_v[k] - \sum_{v \in \mathcal{V}_{train}} \sum_{k=0}^{K-1} Y_v[k] \ln P'_v[k], \tag{11}
$$

where $\mathcal{V}_{train}$ is the set of training nodes, $Y \in \mathbb{R}^{n \times K}$ is the ground truth label matrix, $K$ denotes the number of classes.

### 3.4 Theoretical Analysis

This section provides a theoretical analysis of G$^2$Former, focusing on the filtering properties of the global-aware graph filter and the time complexity.

**Theorem 3.1.** *Consider $\alpha > 0, \beta > 0$, when $\alpha + \beta \to \infty$, $\{\mathcal{F}(\tilde{\mathbf{L}}_S; i, \alpha+\beta-i, \Theta) \mid i \in \{0, .., \alpha+\beta\}\}$ and $\{\mathcal{F}(\tilde{\mathbf{L}}_S; i, \alpha + \beta, \Theta) \mid i \in \{0, .., \alpha+\beta\}\}$ are capable of covering any specific frequency band within the range $(0, 2)$ and $(0, 1]$, respectively.*

The proof can be found in the Appendix A. Theorem 3.1 highlights the strong capability of global-aware graph filters and underscores the importance of the hyperparameter $\alpha + \beta$. An increase in $\alpha + \beta$ leads to greater diversity among filtering channels, which improves the quality of the guidance signals. Additionally, the shared use of the FFNs (i.e., $f_{\theta_1}, f_{\theta_2}$) across more channels imposes stronger regularization on the Transformer and effectively prevents it from overfitting to a limited number

Table 1: Comparison between our model and the baselines over **homophilous graphs** (%). The reported metric is Accuracy.

| | Cora | CiteSeer | PubMed | Computer | Photo | CS | Physics | WikiCS |
|---|---|---|---|---|---|---|---|---|
| GraphGPS | $83.12_{\pm1.15}$ | $72.70_{\pm1.25}$ | $79.96_{\pm0.15}$ | $91.80_{\pm0.55}$ | $94.83_{\pm0.15}$ | $94.00_{\pm0.21}$ | $96.51_{\pm0.20}$ | $78.63_{\pm0.45}$ |
| NAGphormer | $80.35_{\pm1.10}$ | $70.12_{\pm1.50}$ | $80.02_{\pm1.10}$ | $91.65_{\pm0.35}$ | $96.12_{\pm0.17}$ | $95.81_{\pm0.19}$ | $97.33_{\pm0.11}$ | $77.95_{\pm0.86}$ |
| EXphormer | $83.22_{\pm1.35}$ | $71.88_{\pm1.15}$ | $79.65_{\pm0.75}$ | $91.55_{\pm0.22}$ | $95.62_{\pm0.42}$ | $96.00_{\pm0.27}$ | $96.90_{\pm0.13}$ | $79.26_{\pm0.60}$ |
| GOAT | $83.20_{\pm1.22}$ | $72.02_{\pm1.27}$ | $79.60_{\pm0.61}$ | $92.32_{\pm0.42}$ | $94.43_{\pm0.22}$ | $93.72_{\pm0.21}$ | $96.37_{\pm0.25}$ | $77.97_{\pm0.60}$ |
| NodeFormer | $82.12_{\pm0.95}$ | $72.27_{\pm1.21}$ | $80.10_{\pm0.73}$ | $87.12_{\pm0.48}$ | $93.40_{\pm0.65}$ | $95.71_{\pm0.29}$ | $96.45_{\pm0.27}$ | $75.15_{\pm0.99}$ |
| SGFormer | $\mathbf{84.72}_{\pm0.77}$ | $\mathbf{72.70}_{\pm1.10}$ | $\underline{80.63}_{\pm0.44}$ | $92.42_{\pm0.67}$ | $95.60_{\pm0.35}$ | $95.75_{\pm0.25}$ | $96.70_{\pm0.23}$ | $80.10_{\pm0.47}$ |
| Polynormer | $83.33_{\pm0.89}$ | $72.20_{\pm0.95}$ | $79.12_{\pm0.51}$ | $93.76_{\pm0.15}$ | $96.60_{\pm0.25}$ | $95.30_{\pm0.20}$ | $97.20_{\pm0.10}$ | $80.12_{\pm0.85}$ |
| Spexphormer | $83.10_{\pm1.03}$ | $71.86_{\pm0.86}$ | $79.33_{\pm0.64}$ | $91.12_{\pm0.11}$ | $95.35_{\pm0.45}$ | $94.90_{\pm0.17}$ | $96.71_{\pm0.10}$ | $78.12_{\pm0.77}$ |
| GCN | $\underline{84.56}_{\pm0.71}$ | $72.28_{\pm0.49}$ | $\mathbf{81.16}_{\pm0.50}$ | $\underline{93.95}_{\pm0.15}$ | $96.12_{\pm0.50}$ | $96.12_{\pm0.10}$ | $\underline{97.44}_{\pm0.12}$ | $80.33_{\pm0.57}$ |
| GraphSAGE | $83.68_{\pm0.55}$ | $72.21_{\pm0.44}$ | $79.70_{\pm0.64}$ | $93.22_{\pm0.17}$ | $\underline{96.71}_{\pm0.25}$ | $\underline{96.33}_{\pm0.15}$ | $97.10_{\pm0.11}$ | $80.61_{\pm0.38}$ |
| GAT | $84.22_{\pm0.75}$ | $72.22_{\pm0.87}$ | $80.12_{\pm0.72}$ | $94.01_{\pm0.11}$ | $96.63_{\pm0.34}$ | $96.20_{\pm0.13}$ | $97.26_{\pm0.09}$ | $\underline{81.01}_{\pm0.57}$ |
| G²Former | $83.72_{\pm0.55}$ | $\underline{72.62}_{\pm0.63}$ | $79.30_{\pm0.68}$ | $\mathbf{94.29}_{\pm0.10}$ | $\mathbf{97.06}_{\pm0.10}$ | $\mathbf{96.53}_{\pm0.09}$ | $\mathbf{97.60}_{\pm0.05}$ | $\mathbf{81.14}_{\pm0.22}$ |

of channels. Nevertheless, $\alpha + \beta$ should not be excessively large. Beyond the risk of performance degradation due to over-regularization, redundant channels just incur additional computational cost. In line with Occam's Razor [2], the number of channels should be kept minimal yet sufficient. We further discuss it in Appendix B.1.

**Time Complexity.** The main computational cost of G²Former lies in the Transformer-based global graph filtering (i.e., Equation 10) and the downstream GNN encoding. It is important to note that the computation must follow the linear attention computation order described in Equation 1; otherwise, its complexity would increase to $\mathcal{O}(|\mathcal{V}|^2)$. For example, $\tilde{\mathbf{L}}_S f_{\theta_2}(\mathbf{X})$ is equal to:

$$f_{\theta_2}(\mathbf{X}) - \mathbf{D}_S^{-1/2}\mathbf{A}_S\mathbf{D}_S^{-1/2}f_{\theta_2}(\mathbf{X}) = f_{\theta_2}(\mathbf{X}) - \mathbf{D}_S^{-1/2}(\mathbf{Q}'(\mathbf{K}'^T(\mathbf{D}_S^{-1/2}f_{\theta_2}(\mathbf{X})))), \qquad (12)$$

where $\mathbf{D}_S = \text{diag}(\mathbf{Q}'(\mathbf{K}'^T\mathbf{1}))$. So global graph filtering only needs $\mathcal{O}((\alpha + \beta)|\mathcal{V}|d'^2)$. The complexity of the GNN is $\mathcal{O}(|\mathcal{E}|)$, and with the additional cost of the feed-forward network (FNN) being $\mathcal{O}(|\mathcal{V}|dd')$, the overall complexity of G²Former is $\mathcal{O}(|\mathcal{V}|((\alpha + \beta)d'^2 + dd') + |\mathcal{E}|)$. Hence, G²Former still has linear complexity with respect to the number of nodes and edges.

## 4 Experiments

We conducted extensive evaluation experiments on three types different real-world graph datasets (including homophilous, heterophilous and large graphs) to verify the effectiveness of G²Former. Specifically, we first evaluate the performance of G²Former on these node classification datasets, addressing the question of *whether Transformers can benefit GNNs*. We conducted ablation studies on two types of channel initialization strategies and global attention module (including test error and visualization). In addition, we perform a sensitivity analysis on the number of filtering channels in Appendix B.1. Finally, we report the training and inference cost of G²Former to demonstrate its scalability in Appendix B.2.

**Experimental Setup.** Appendix C.1 provides detailed statistics and splitting of the 17 standard datasets (8 homophilous, 6 heterophilous and 3 large graphs). Note that for heterophilous datasets, we use the new version [46] of Chameleon and Squirrel, which removes overlapped nodes. The baselines mainly include **8 state-of-the-art GTs** (GraphGPS [47], NAGphormer [7], EXphormer [52], GOAT [27], NodeFormer [62], SGFormer [63], Polynormer [13] and Spexphormer [53]) and the three **well-tuned classic GNNs**: GCN [26], GraphSAGE [20] and GAT [57]. Detailed descriptions and hyperparameter configurations can be found in the Appendix C.1, C.2. We use the official implementations or public benchmarks [36] of baselines whenever possible. We follow [36] or their configurations to tune the hyperparameters for better performance. We report the mean and standard deviation over five independent runs. The experimental environment details and the specific hyperparameter settings for G²Former are also provided in the Appendix C.3 and Appendix D.

### 4.1 Performance on Diverse Graphs

Performance comparisons on homophilous, heterophilous, and large-scale graphs are presented in Tables 1, 2 and 3, respectively. The best and second-best results are highlighted in **bold**

Table 2: Comparison between our model and the baselines over **heterophilous graphs** (%). The reported metric is ROC-AUC for the Minesweeper and Questions and Accuracy for all others.

| | Squirrel | Chameleon | Amazon-Ratings | Roman-Empire | Minesweeper | Questions |
|---|---|---|---|---|---|---|
| GraphGPS | $39.68_{\pm 2.69}$ | $41.23_{\pm 3.89}$ | $53.33_{\pm 0.69}$ | $82.45_{\pm 0.67}$ | $90.66_{\pm 0.89}$ | $72.33_{\pm 1.32}$ |
| NAGphormer | $39.33_{\pm 1.20}$ | $41.72_{\pm 4.61}$ | $51.42_{\pm 0.68}$ | $74.49_{\pm 0.66}$ | $85.03_{\pm 0.77}$ | $68.45_{\pm 1.23}$ |
| EXphormer | $39.80_{\pm 1.67}$ | $39.13_{\pm 3.16}$ | $53.34_{\pm 0.38}$ | $89.13_{\pm 0.41}$ | $90.38_{\pm 0.63}$ | $74.01_{\pm 1.10}$ |
| GOAT | $35.05_{\pm 1.29}$ | $35.06_{\pm 5.77}$ | $45.11_{\pm 0.59}$ | $71.88_{\pm 0.93}$ | $81.88_{\pm 1.33}$ | $75.80_{\pm 1.71}$ |
| NodeFormer | $38.77_{\pm 2.67}$ | $36.33_{\pm 4.09}$ | $43.80_{\pm 0.59}$ | $74.90_{\pm 0.85}$ | $87.73_{\pm 0.67}$ | $74.98_{\pm 1.58}$ |
| SGFormer | $42.60_{\pm 2.40}$ | $44.32_{\pm 3.91}$ | $54.04_{\pm 0.60}$ | $80.10_{\pm 0.49}$ | $91.43_{\pm 0.45}$ | $73.83_{\pm 0.69}$ |
| Polynormer | $42.05_{\pm 2.37}$ | $41.88_{\pm 3.88}$ | $54.96_{\pm 0.29}$ | $\underline{92.56}_{\pm 0.35}$ | $97.50_{\pm 0.49}$ | $78.91_{\pm 0.80}$ |
| Spexphormer | $40.33_{\pm 2.79}$ | $40.30_{\pm 3.19}$ | $53.27_{\pm 0.77}$ | $83.03_{\pm 0.69}$ | $90.73_{\pm 0.23}$ | $73.02_{\pm 0.76}$ |
| GCN | $\underline{45.03}_{\pm 1.68}$ | $\mathbf{44.91}_{\pm 4.57}$ | $53.82_{\pm 0.76}$ | $91.23_{\pm 0.22}$ | $\underline{97.79}_{\pm 0.28}$ | $\underline{78.92}_{\pm 0.62}$ |
| GraphSAGE | $41.33_{\pm 1.62}$ | $\underline{44.80}_{\pm 4.53}$ | $55.14_{\pm 0.23}$ | $91.02_{\pm 0.21}$ | $97.73_{\pm 0.52}$ | $77.20_{\pm 1.32}$ |
| GAT | $41.81_{\pm 2.16}$ | $44.22_{\pm 4.29}$ | $\underline{55.16}_{\pm 0.21}$ | $90.41_{\pm 0.17}$ | $97.75_{\pm 0.93}$ | $77.90_{\pm 0.59}$ |
| G²Former | $\mathbf{45.53}_{\pm 1.76}$ | $44.34_{\pm 4.39}$ | $\mathbf{55.86}_{\pm 0.13}$ | $\mathbf{93.03}_{\pm 0.33}$ | $\mathbf{99.45}_{\pm 0.12}$ | $\mathbf{79.55}_{\pm 0.31}$ |

and underlined, respectively. **On homophilous graphs**, G²Former improves the generalization performance of the best-performing GNNs by 0.16%–0.36%, and that of the best-performing Graph Transformers by 0.28%–1.27%. GNNs generally perform well on homophilous graphs, which can be attributed to the compatibility between their low-pass filtering nature and homophily. Notably, SGFormer achieves the best performance on Cora, Citeseer, and PubMed, but performs poorly on other real-world datasets. In contrast, Polynormer does not overfit to these three toy datasets and demonstrates relatively strong applicability among existing GTs. **On heterophilous graphs**, G²Former yields generalization improvements ranging from 0.51% to 1.7% over the best-performing GNNs, and from 0.51% to 6.88% over the best-performing GTs.

The advantage of G²Former is particularly evident in heterophile graphs, which typically exhibit more complex connectivity patterns - where the labels of neighboring nodes often differ from those of the central node. The global graph filter in G²Former exhibits band-pass characteristics, enabling it to capture both low and high-frequency components on heterophilous graphs. In contrast, both GNNs and existing GTs are single-channel (i.e., low-pass only) and thus incapable of effectively capturing high-frequency signals. **For large graphs**, G²Former provides generalization improvements ranging from 0.2% to 0.6% over the best-performing GNNs, and from 0.04% to 0.9% over the best-performing GTs. Its advantage remains more pronounced on heterophilous graphs. As the scale of heterophilous graphs increases, the performance gap among GTs also widens, which is closely related to the design of their attention modules. In such scenarios, the introduction of overly complex mechanisms may limit its applicability. Overall, G²Former contributes to better generalization in GNNs, yet it falls short of surpassing the inherent performance ceiling imposed by GNNs.

Table 3: Comparison between our model and the baselines over **large graphs** (%). The reported metric is Accuracy. OOM means out of memory.

| | ogbn-arxiv | ogbn-products | pokec |
|---|---|---|---|
| Type | Homophilous | Homophilous | Heterophilous |
| GraphGPS | $71.10_{\pm 0.50}$ | OOM | OOM |
| NAGphormer | $70.90_{\pm 0.29}$ | $74.33_{\pm 0.25}$ | $76.10_{\pm 0.65}$ |
| EXphormer | $72.11_{\pm 0.35}$ | OOM | OOM |
| GOAT | $72.81_{\pm 0.31}$ | $82.29_{\pm 0.66}$ | $72.50_{\pm 0.77}$ |
| NodeFormer | $67.66_{\pm 0.26}$ | $74.10_{\pm 0.53}$ | $71.00_{\pm 1.30}$ |
| SGFormer | $72.42_{\pm 0.42}$ | $81.66_{\pm 0.54}$ | $82.33_{\pm 0.71}$ |
| Polynormer | $\underline{73.33}_{\pm 0.28}$ | $\underline{83.61}_{\pm 0.15}$ | $86.01_{\pm 0.20}$ |
| Spexphormer | $70.84_{\pm 0.33}$ | $82.10_{\pm 0.50}$ | $74.80_{\pm 0.15}$ |
| GCN | $73.31_{\pm 0.32}$ | $82.35_{\pm 0.20}$ | $\underline{86.31}_{\pm 0.20}$ |
| GraphSAGE | $73.05_{\pm 0.33}$ | $83.47_{\pm 0.43}$ | $85.90_{\pm 0.23}$ |
| GAT | $73.30_{\pm 0.23}$ | $80.90_{\pm 0.20}$ | $86.12_{\pm 0.22}$ |
| G²Former | $\mathbf{73.63}_{\pm 0.23}$ | $\mathbf{83.98}_{\pm 0.03}$ | $\mathbf{86.81}_{\pm 0.06}$ |

## 4.2 Test-time Behavior of Attention Mechanism

This section investigates the encoding capacity of attention modules within GTs, this aspect that has been largely overlooked in prior work. To reveal the inherent differences, we remove the GNN component and train only the attention module for encoding. We compare the attention mechanisms of two top-performing GTs—SGFormer and Polynormer—and report their test errors after training, as shown in Figure 4. In Figure 4b and Figure 4d, the regularization effect of G²Former's attention is evident, effectively suppressing test error oscillations. Even in more stable cases, G²Former's attention consistently maintains the lowest loss levels. This experiment demonstrates that under

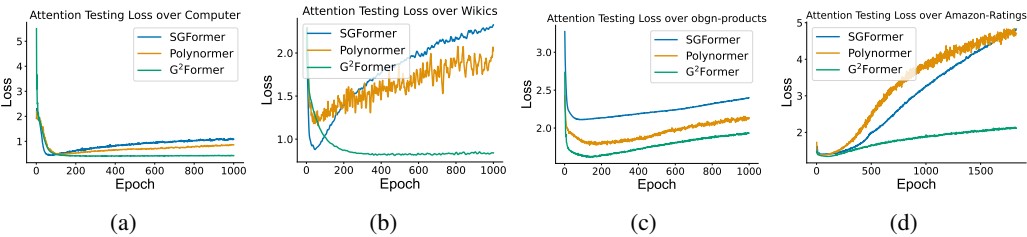

(a)           (b)           (c)           (d)

Figure 4: A comparison of the test error behavior of the attention mechanism in $G^2$Former versus those in the other two GTs.

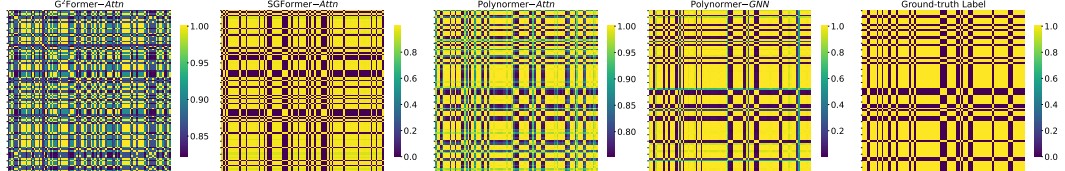

Figure 5: Visualization of the attention mechanisms of the $G^2$Former, SGFormer, and Polynormer, along with Polynormer-GNN and the corresponding label matrix.

multi-channel constraints, $G^2$Former's attention can adapt to various types and scales of graphs, effectively avoiding the introduction of excessive detrimental topological noise into the GNN.

### 4.3 Channel Initialization Strategies

This section investigates the differences between the two channel initialization strategies (full-spectrum and low-spectrum) described in Section 3.1, with the results presented in Figure 6. The dashed line indicates the performance of the best-performing GNN. Performance degradation is observed on 11 out of the 12 datasets, with the exception of ogbn-arxiv. Moreover, on four homophilous graphs — CS, Physics and WikiCS — the results are even worse than those of the best-performing GNN. This provides strong evidence for the necessity of high-frequency channel constraints. If the attention module only covers the low-frequency band, it may still exhibit "laziness" in learning. The inclusion of high-frequency components helps to capture edge patterns (i.e., heterophilous connections), thereby mitigating the influence of trivial or spurious connections learned by the module. For heterophilous graphs, the target signal is relatively dispersed across the frequency spectrum and less concentrated in the

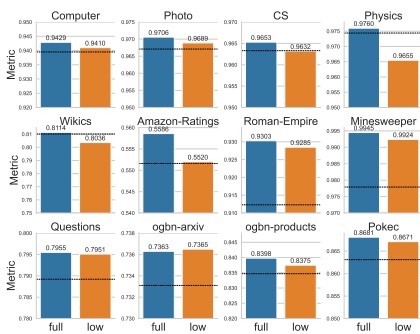

Figure 6: Performance under different channel initialization strategies.

low-frequency band [3, 9]. This spectral dispersion makes the low-spectrum filter bank less prone to overfitting, but the inability to capture components that reside in the high-frequency range still limits its performance.

### 4.4 Visualization

To further investigate the role of attention mechanisms in GTs, we randomly sample 100 nodes on Minesweeper to visualize them in Figure 5. Compared to the label matrix, the node representation similarity produced by Polynormer-GNN aligns closely with the ground truth. Despite the inclusion of an attention module downstream, no performance gain over the GNN is observed. And the close resemblance between Polynormer-Attn and Polynormer-GNN further indicates that its attention layer mainly repeats the already captured features. In contrast, SGFormer introduces substantial topological noise through its attention mechanism. In $G^2$Former, the multi-channel constraint plays a crucial role:

spurious connections are effectively suppressed, while falsely missing edges still receive relatively high attention scores. As observed in the highlighted regions, this sparsity effect is reminiscent of Lasso Regression [55]. G$^2$Former tends to emphasize the intersection of above different attention mechanisms, thereby guiding the downstream GNN to focus more on essential features.

## 5 Conclusions

This paper clarifies that Transformers do not empirically provide a universal performance boost for Graph Neural Networks, but are better suited as a form of data augmentation to enhance generalization. We introduce a global-aware graph filtering framework with multi-channel constraints that stabilizes attention, promotes feature focus, and reduces overfitting. Our findings highlight the value of revisiting the intrinsic structure of graphs over increasingly complex network architectures.

## Acknowledgments and Disclosure of Funding

This work was supported by the National Key Research and Development Program of China (No.2023YFB3307203).

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

# A   The Proof of Theorem 3.1.

*Proof.* In the following content, we abbreviate the two channel initialization strategies $\{\mathcal{F}(\tilde{\mathbf{L}}_S; i, \alpha + \beta - i, \Theta) \mid i \in \{0, .., \alpha + \beta\}\}$ and $\{\mathcal{F}(\tilde{\mathbf{L}}_S; i, \alpha + \beta, \Theta) \mid i \in \{0, .., \alpha + \beta\}\}$ as $\mathcal{F}_{full}$ and $\mathcal{F}_{low}$, respectively. Each filter $\mathcal{F}(\tilde{\mathbf{L}}_S; \alpha, \beta, \Theta)$ among them, its frequency response function is $\frac{1}{C}g_S(\lambda) = \frac{1}{C}\lambda^\alpha(2 - \lambda)^\beta$, $C$ corresponds to Equation 6. Consider $\alpha > 0, \beta > 0, \lambda \in [0, 2]$, $\Lambda \sim g_S(\lambda)$, we use the mean $\hat{\lambda} = \mathbb{E}(\Lambda)$ to represent the band-pass frequency of $g_S(\lambda)$ like Figure 3.

For $\mathcal{F}_{full}$, $\mathbb{E}(\Lambda)$ is written as:

$$
\begin{aligned}
\mathbb{E}(\Lambda) &= \int_0^2 \lambda \cdot g_S(\lambda)\mathrm{d}\lambda = \int_0^2 \lambda \cdot \frac{1}{C}\lambda^\alpha(2 - \lambda)^\beta\mathrm{d}\lambda \\
&= \int_0^2 \frac{1}{C}\lambda^{\alpha+1}(2 - \lambda)^\beta\mathrm{d}\lambda.
\end{aligned}
\tag{13}
$$

According to Equation 4, 5, 6, we can get:

$$
\begin{aligned}
\mathbb{E}(\Lambda) &= \frac{1}{C}\int_0^2 \lambda^{\alpha+1}(2 - \lambda)^\beta\mathrm{d}\lambda \\
&= \frac{\Gamma(\alpha + \beta + 2)}{\Gamma(\alpha + 1)\Gamma(\beta + 1)2^{\alpha+\beta+1}} \cdot 2^{\alpha+\beta+2} \cdot \frac{\Gamma(\alpha + 2)\Gamma(\beta + 1)}{\Gamma(\alpha + \beta + 3)} \\
&= \frac{2(\alpha + 1)}{\alpha + \beta + 2}.
\end{aligned}
\tag{14}
$$

Similarly, the mean $\mathbb{E}(\Lambda^2)$ can be written as:

$$
\begin{aligned}
\mathbb{E}(\Lambda^2) &= \frac{1}{C}\int_0^2 \lambda^{\alpha+2}(2 - \lambda)^\beta\mathrm{d}\lambda \\
&= \frac{\Gamma(\alpha + \beta + 2)}{\Gamma(\alpha + 1)\Gamma(\beta + 1)2^{\alpha+\beta+1}} \cdot 2^{\alpha+\beta+3} \cdot \frac{\Gamma(\alpha + 3)\Gamma(\beta + 1)}{\Gamma(\alpha + \beta + 4)} \\
&= \frac{4(\alpha + 1)(\alpha + 2)}{(\alpha + \beta + 2)(\alpha + \beta + 3)}.
\end{aligned}
\tag{15}
$$

The variance $\mathrm{Var}(\Lambda)$ can be written as:

$$
\begin{aligned}
\mathrm{Var}(\Lambda) &= \mathbb{E}(\Lambda^2) - (\mathbb{E}(\Lambda))^2 \\
&= \frac{4(\alpha + 1)(\alpha + 2)}{(\alpha + \beta + 2)(\alpha + \beta + 3)} - \left(\frac{2(\alpha + 1)}{\alpha + \beta + 2}\right)^2 \\
&= \frac{4(\alpha + 1)(\beta + 1)}{(\alpha + \beta + 2)^2(\alpha + \beta + 3)}.
\end{aligned}
\tag{16}
$$

If $\alpha + \beta \to \infty$, $\mathrm{Var}(\Lambda) = \frac{4(\alpha+1)(\beta+1)}{(\alpha+\beta+2)^2(\alpha+\beta+3)} \to 0$, $\mathbb{E}(\Lambda) = \frac{2(\alpha+1)}{\alpha+\beta+2} \to \frac{2\alpha}{\alpha+\beta} = \frac{2\frac{\alpha}{\beta}}{\frac{\alpha}{\beta}+1}$. Let $x = \frac{\alpha}{\beta} \in (0, +\infty)$, $\mathbb{E}(\Lambda) \to \frac{2x}{x+1}$. It means that $\mathcal{F}_{full}$ can concentrate on any $\mathbb{E}(\Lambda) \in (0, 2)$, which completes the proof of the frequency band coverage property of $\mathcal{F}_{full}$.

For $\mathcal{F}_{low}$, its coverage within the interval $(0, 1)$ can be referred to $\mathcal{F}_{\text{full}}$. For the last filter $\{\mathcal{F}(\tilde{\mathbf{L}}_S; \alpha + \beta, \alpha + \beta, \Theta)$ in $\mathcal{F}_{low}$, where $x = 1$, we have $\mathbb{E}(\Lambda) = 1$, the band-pass frequency of $\mathcal{F}_{\text{low}}$ reaches its maximum. Hence, the frequency band coverage range of $\mathcal{F}_{\text{low}}$ is $(0, 1]$.

$\square$

# B   Additional Experiments

## B.1   Sensitivity

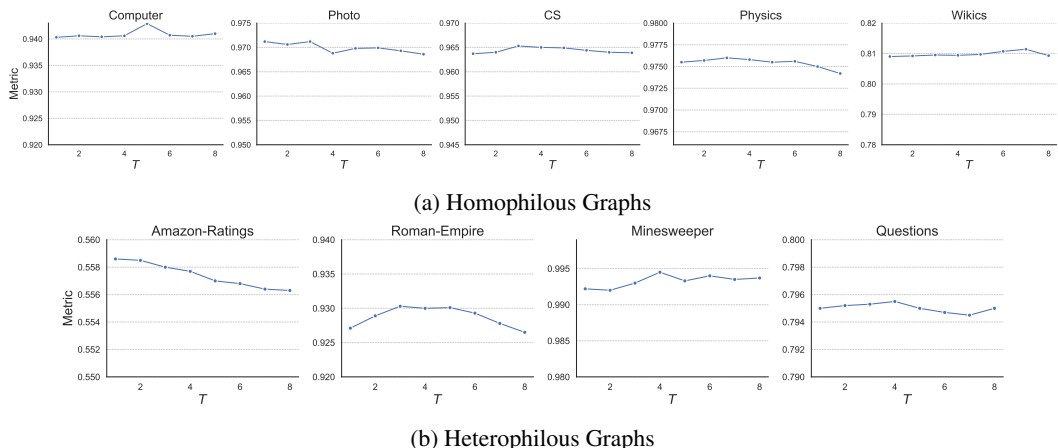

(a) Homophilous Graphs

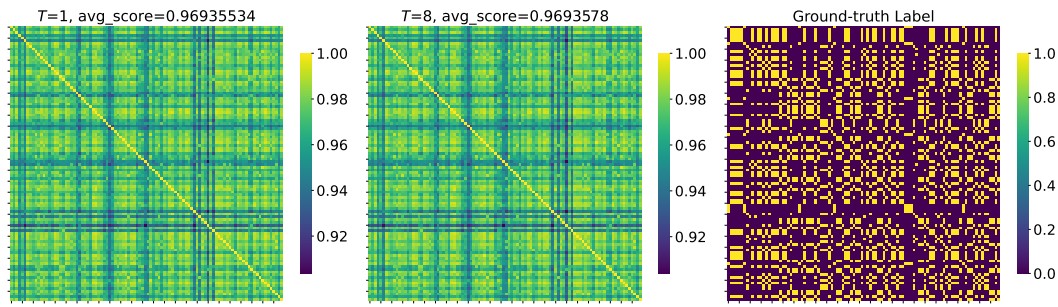

(b) Heterophilous Graphs

Figure 7: Effect of varying $T$ on model performance, $T = \alpha + \beta$.

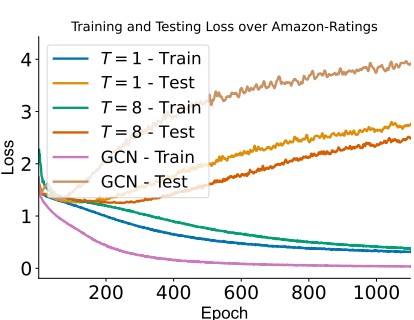

Figure 8: Visualization of the attention mechanisms of the G²Former, SGFormer, and Polynormer, along with Polynormer-GNN and the corresponding label matrix.

As a critical hyperparameter in G²Former, $T = \alpha + \beta$ directly controls the extent to which the attention mechanism is constrained. To assess its impact, we performed a sensitivity analysis, with the experimental results shown in Figure 7. In conjunction with Table 12, the optimal number of filters tends to concentrate between 2 and 4 across both homophilous and heterophilous graphs. The maximum number of filters is used in homophilous graphs (e.g., Wikics, $T$=7), while the minimum appears in heterophilous ones (e.g., Amazon-Ratings, $T$=1). A possible explanation is that, in homophilous graphs, due to the abundance of homophilous connections, single-channel attention mechanisms (i.e., low-pass filters) tend to induce nearly fully connected graphs, introducing substantial topological noise—particularly when the number of

Figure 9: Training and Testing Loss over Amazon-Ratings under different $T$.

classes is large. Larger $T$ impose stronger constraints, which help suppress spurious connections. Heterophilous graphs present more complex challenges, often requiring manual tuning to apply appropriate levels of constraint. For more difficult datasets (e.g., Amazon-Ratings), overly strong constraints may hinder the downstream GNN from effectively focusing on the most relevant features.

Taking Amazon-Ratings as an example, Figure 8 illustrates that increasing $T$ from 1 to 8 leads to minimal changes in the attention module, indicating the presence of redundant channels. Figure 9 presents the training and testing losses of G²Former under $T$=1 and $T$=8. As shown, while $T$=8 achieves the most stable test loss, the performance begins to fluctuate (see Figure 7b), suggesting that the constraint may be overly strong. Specifically, when $T$ becomes too large, $f_{\theta_2}$ also must update parameters across a greater number of channels, thus increasing the optimization difficulty. In

Table 4: Runtime/Epoch (including training, validation, and testing) on large graphs (s).

| Dataset | ogbn-arxiv | ogbn-products | pokec |
|---|---|---|---|
| SGFormer | 0.33 | 11.13 | 7.43 |
| Polynormer | 1.33 | 15.06 | 7.24 |
| $G^2$Former | 1.39 | 10.63 | 6.04 |

Table 5: Performance of band-pass filters under different heterophilous levels.

| Dataset | Photo | ogbn-products | pokec | Minesweeper |
|---|---|---|---|---|
| Hete. | 0.1674 | 0.1887 | 0.5552 | 0.2815 |
| $\mathbf{X}$ | 0.5466(0.07/0.06) | 0.5043(37.80/37.16) | 0.5299(27.56/24.45) | 0.5544(0.97/0.78) |
| $\mathcal{F}_0$ | 0.6835(3.27/1.52) | 0.4895(0.58/0.60) | 0.5311(1.02/0.90) | 0.5855(0.25/0.18) |
| $\mathcal{F}_1$ | 0.6835(6.55/3.03) | 0.4892(1.72/1.80) | 0.5310(3.02/2.67) | 0.5855(0.96/0.68) |
| $\mathcal{F}_2$ | 0.6835(3.27/1.52) | 0.4891(1.72/1.80) | 0.5310(2.98/2.64) | 0.5864(1.40/0.98) |
| $\mathcal{F}_3$ | - | 0.4889(0.57/0.60) | 0.5309(0.98/0.87) | 0.5873(0.90/0.63) |
| $\mathcal{F}_4$ | - | - | - | 0.5882(0.22/0.15) |
| $\mathbf{X}'$ | 0.6698(12.36/6.09) | 0.50(25.82/25.82) | 0.5288(14.69/13.09) | 0.5835(2.94/2.10) |

contrast, with $T$=1, the test loss lies between that of $T$=8 and tuned-GCN, indicating a more suitable level of regularization.

## B.2 Scalability

Table 4 reports the runtime per epoch (including training, validation, and testing) of $G^2$Former and the other two SOTA linear GTs. Note that full-batch training is used for ogbn-arxiv, while the remaining two datasets share the same batch size setting. The results indicate that $G^2$ Former incurs a computational overhead comparable to linear GTs, which is consistent with the complexity analysis in Section 3.4. The GNN backbone of $G^2$Former relies solely on efficient operations supported by modern graph learning frameworks (e.g., PyG, DGL), while the global-aware filter consists only of linear matrix multiplications. Therefore, we believe $G^2$Former to be highly efficient.

## B.3 Effect of Graph-Guided Filtering

In Table 5, we present the response of different spectral channels under varying levels of heterophily, where heterophily (Hete.) is defined as the proportion of heterophilous edges among all edges. The table reports the $Dis(Dis_{hete}/Dis_{homo})$, for $\mathbf{X}$ and $\mathbf{X}'$, representing the features before and after graph-guided filtering, respectively. $\mathcal{F}_i$ denotes the filter associated with the $i$-th channel, where a larger $i$ corresponds to higher-frequency bands. As shown in the table, for smaller-scale graphs such as Photo and Minesweeper, graph-guided filtering tends to enhance the distinction between heterophilous node pairs, resulting in increased $Dis$ scores in $\mathbf{X}'$. In contrast, for large-scale graphs such as ogbn-products and pokec, the filtering operation exhibits a compressive effect on node attributes, reducing both $Dis\_hete$ and $Dis\_homo$. Nevertheless, the overall feature distribution remains close to the original, as reflected by similar $Dis$ scores before and after filtering. These findings highlight the adaptive role of graph-guided filtering: enhancing local discriminability in smaller graphs, while promoting feature compactness in larger graphs without significantly distorting the global structure.

## B.4 Results in Comparison with Heterophilous and Spectral GNNs

In addition to H2GCN [71] and FAGCN [3], we also provide experimental results (In Table 6, our method consistently demonstrates superior performance across various benchmarks.) of CPGNN [72], GPRGNN [9], FSGNN [38], and GloGNN [31]. We also provide a comparison between our method and these spectral GNNs (BernNet [22], JacobiConv [59], OptBasisGNN [19] and UniFilter [24]), as shown in Table 7, 8 below. The best performance is highlighted in bold. Our method still demonstrates superior performance.

Table 6: Comparison of model performance on heterophilous graphs.

| Dataset | Squirrel | Chameleon | Amazon-Ratings | Roman-Empire | Minesweeper | Questions |
|---------|----------|-----------|----------------|--------------|-------------|-----------|
| Metric | Accuracy | Accuracy | Accuracy | Accuracy | ROC-AUC | ROC-AUC |
| H2GCN | $34.75_{\pm 0.97}$ | $27.80_{\pm 3.87}$ | $36.40_{\pm 0.29}$ | $60.01_{\pm 0.48}$ | $88.65_{\pm 0.37}$ | $62.37_{\pm 1.13}$ |
| FAGCN | $36.33_{\pm 1.62}$ | $34.67_{\pm 3.19}$ | $40.28_{\pm 0.68}$ | $62.89_{\pm 0.77}$ | $50.19_{\pm 1.34}$ | $67.99_{\pm 1.51}$ |
| CPGNN | $30.01_{\pm 2.11}$ | $32.75_{\pm 3.79}$ | $38.66_{\pm 0.62}$ | $62.37_{\pm 0.44}$ | $52.01_{\pm 0.73}$ | $65.97_{\pm 1.67}$ |
| GPRGNN | $39.01_{\pm 2.06}$ | $39.64_{\pm 3.83}$ | $45.02_{\pm 0.33}$ | $65.13_{\pm 0.71}$ | $85.26_{\pm 0.58}$ | $54.38_{\pm 1.49}$ |
| FSGNN | $34.99_{\pm 1.27}$ | $40.33_{\pm 3.59}$ | $51.87_{\pm 0.81}$ | $79.65_{\pm 0.41}$ | $90.11_{\pm 0.77}$ | $78.85_{\pm 0.92}$ |
| GloGNN | $35.36_{\pm 2.14}$ | $23.99_{\pm 4.37}$ | $36.97_{\pm 0.17}$ | $60.21_{\pm 0.67}$ | $51.26_{\pm 1.12}$ | $65.32_{\pm 1.28}$ |
| Ours | $\mathbf{45.53}_{\pm 1.76}$ | $\mathbf{44.34}_{\pm 4.39}$ | $\mathbf{55.86}_{\pm 0.13}$ | $\mathbf{93.03}_{\pm 0.33}$ | $\mathbf{99.45}_{\pm 0.12}$ | $\mathbf{79.55}_{\pm 0.31}$ |

Table 7: Comparison of model performance on homophilous graphs.

| | Computer | Photo | cs | physics | wikics |
|---|----------|-------|-----|---------|--------|
| BernNet | 92.63±0.5 | 94.65±0.2 | 94.77±0.3 | 96.54±0.1 | 75.29±0.5 |
| JacobiConv | 92.32±0.2 | 93.76±0.1 | 94.21±0.5 | 96.17±0.1 | 75.47±1.1 |
| OptBasisGNN | 91.04±0.4 | 95.31±0.3 | 95.66±0.6 | 96.81±0.1 | 77.63±0.9 |
| UniFilter | 93.20±0.3 | 93.98±0.4 | 93.22±0.1 | 96.77±0.3 | 78.82±0.7 |
| G$^2$Former | **94.29**±0.1 | **97.06**±0.1 | **96.53**±0.1 | **97.60**±0.1 | **81.14**±0.2 |

## B.5 Comparison of Generalization Gap

We report the model's generalization gap [65] in Table 9, 10 below. The tuned GNN refers to the classical GNN that achieves the highest performance score on the corresponding dataset. Our method consistently achieves the lowest generalization gap.

## C Dataset and Baselines Details

### C.1 Dataset

Table 11 presents detailed statistics of the 17 datasets used in our experiments. For the training/validation/test splits, we directly adopt the splits from [36] (0.6/0.2/0.2) for Computer, Photo, CS, and Physics. Squirrel and Chameleon follow the version of [46], which removes duplicate nodes. The remaining datasets use the official splits [32, 23, 39].

Specifically, **Cora**, **CiteSeer**, and **PubMed** [50] are widely used toy citation networks. **Computer** and **Photo** [51] are co-purchase networks where nodes represent products and edges indicate frequently co-purchased items. **CS** and **Physics** [51] are co-authorship networks, with nodes denoting authors and edges representing collaborations. **WikiCS** [39] is a citation network of computer science papers.

**Squirrel** and **Chameleon** [48] are two widely studied Wikipedia page networks centered around specific topical domains. Other 4 heterophilous datasets [46]: **Roman-Empire** is a word-level graph constructed from the Roman Empire Wikipedia article, where nodes represent individual words and edges link words that are either sequential or syntactically related. **Amazon-Ratings** represents a product co-purchasing network, with nodes as products and edges connecting items frequently bought

Table 8: Comparison of model performance on heterophilous graphs.

| | amazon_ratings | roman_empire | minesweeper | questions |
|---|----------------|--------------|-------------|-----------|
| BernNet | 50.70±0.6 | 88.71±0.7 | 89.76±0.5 | 76.04±0.7 |
| JacobiConv | 52.39±0.4 | 89.32±0.3 | 91.62±0.3 | 76.11±1.0 |
| OptBasisGNN | 52.11±0.4 | 88.94±0.6 | 93.50±1.2 | 77.23±0.8 |
| UniFilter | 53.66±0.7 | 91.32±0.6 | 92.38±0.7 | 77.56±0.9 |
| G$^2$Former | **55.86**±0.1 | **93.03**±0.3 | **99.45**±0.1 | **79.55**±0.3 |

Table 9: Generalization gap on homophilous graphs.

| | Computer | Photo | cs | physics | wikics |
|---|---|---|---|---|---|
| SGFormer | 0.2650 | 0.1356 | 0.0963 | 0.0719 | 0.9439 |
| Polynormer | 0.1454 | 0.1622 | 0.0574 | 0.0911 | 0.8236 |
| tuned GNN | 0.0893(GAT) | 0.2341(GraphSAGE) | 0.0874(GraphSAGE) | 0.0089(GCN) | 0.4402(GAT) |
| G$^2$Former | 0.0686↓ | 0.0053↓ | 0.0196↓ | 0.0051↓ | 0.3657↓ |

Table 10: Generalization gap on heterophilous graphs.

| | amazon_ratings | roman_empire | minesweeper | questions |
|---|---|---|---|---|
| SGFormer | 8.2326 | 0.8225 | 0.3561 | 1.7627 |
| Polynormer | 3.3355 | 0.6561 | 0.3393 | 0.7404 |
| tuned GNN | 4.0764(GAT) | 0.6327(GCN) | 0.3475(GCN) | 0.3036(GCN) |
| G$^2$Former | 2.0837↓ | 0.4242↓ | 0.0900↓ | 0.2411↓ |

together. **Minesweeper** is a synthetic dataset, modeling a $100 \times 100$ grid where nodes are grid cells and edges connect adjacent cells. **Questions** is a user interaction graph derived from the Yandex Q question-answering platform, where nodes denote users and edges indicate interactions via answered questions.

**ogbn-arxiv** and **ogbn-products** [23] are datasets released by the Open Graph Benchmark (OGB), representing a citation network of academic papers and an Amazon co-purchasing network, respectively. **Pokec** [29] is a large-scale social network dataset.

## C.2 Baselines

For all datasets, our baselines fall into two main categories: Graph Transformers and well-tuned classical GNNs (GCN, GraphSAGE, and GAT). The Graph Transformer group includes eight models: GraphGPS, NAGphormer, EXphormer, GOAT, NodeFormer, SGFormer, Polynormer, and Spexphormer.

Table 11: Summary of Dataset Statistics for Node Classification.

| Dataset | Type | # Nodes | # Edges | # Feature | Classes |
|---|---|---|---|---|---|
| Cora | Homophilous | 2,708 | 5,278 | 1,433 | 7 |
| Citeseer | Homophilous | 3,327 | 4,522 | 3,703 | 6 |
| Pubmed | Homophilous | 19,717 | 44,324 | 500 | 3 |
| Computer | Homophilous | 13,752 | 245,861 | 767 | 10 |
| Photo | Homophilous | 7,650 | 119,081 | 745 | 8 |
| CS | Homophilous | 18,333 | 81,894 | 6,805 | 15 |
| Physics | Homophilous | 34,493 | 247,962 | 8,415 | 5 |
| Wikics | Homophilous | 11,701 | 216,123 | 300 | 10 |
| Squirrel | Heterophilous | 2,223 | 46,998 | 2,089 | 5 |
| Chameleon | Heterophilous | 890 | 8,854 | 2,325 | 5 |
| Roman-Empire | Heterophilous | 22,662 | 32,927 | 300 | 18 |
| Amazon-Ratings | Heterophilous | 24,492 | 93,050 | 300 | 5 |
| Minesweeper | Heterophilous | 10,000 | 39,402 | 7 | 2 |
| Questions | Heterophilous | 48,921 | 153,540 | 301 | 2 |
| ogbn-arxiv | Homophilous | 169,343 | 1,166,243 | 128 | 40 |
| ogbn-products | Homophilous | 2,449,029 | 61,859,140 | 100 | 47 |
| Pokec | Heterophilous | 1,632,803 | 30,622,564 | 65 | 2 |

For GCN, GraphSAGE, and GAT, we directly use the implementation[2] from [36], where the key hyperparameters for these classical GNNs are already properly configured. For the 8 Graph Transformers, we use their official implementations and tune the critical hyperparameters according to the following configuration:

**GraphGPS.**[3] We perform hyperparameter tuning on the dropout rate from $\{0.1, 0.3, 0.5, 0.7\}$, MPNN layer type within $\{GCN, GraphSAGE, GAT\}$, hidden size from $\{64, 128, 256\}$, the number of layer from $\{1, 2, 3, 4, 5, 6, 7, 8, 9, 10\}$ and attention heads from $\{1, 2, 4\}$. We set the learning rate 0.001 and Performer as the global attention layer type.

**NAGphormer.**[4] We set the learning rate to 0.001 and the hidden size to 512. And We perform hyperparameter tuning on the number of global layers from $\{1, 2, 3, 4, 5, 6, 7, 8, 9, 10\}$, the number of hops from $\{3, 7, 10\}$, dropout rate from $\{0.1, 0.3, 0.5, 0.7\}$ and epochs from $\{1000, 1500, 2500\}$.

**EXphormer.**[5] We choose $\{GCN, GraphSAGE, GAT\}$ as the local model and Exphormer as the global model. We set the learning rate to 0.001. We perform hyperparameter tuning on the hidden size from $\{64, 256, 512\}$, the dropout rate from $\{0.1, 0.3, 0.5, 0.7\}$, the number of layers from $\{1, 2, 3, 4, 5, 6, 7, 8, 9, 10\}$ and the heads of attention from $\{1, 2, 4\}$ and the epochs from $\{1500, 2500\}$.

**GOAT.**[6] We set the "conv type" to "full", the learning rate to 0.001 and the epochs to 2000. We perform hyperparameter tuning on the number of centroids from $\{1024, 2048, 4096\}$, dropout rate from $\{0.1, 0.3, 0.5, 0.7\}$, the hidden size from $\{64, 256, 512\}$, the heads of attention from $\{1, 2, 4\}$ and the number of layers from $\{1, 2, 3, 4, 5, 6, 7, 8, 9, 10\}$.

**NodeFormer.**[7] We set the learning rate to 0.001. We perform hyperparameter tuning on dropout rate from $\{0.1, 0.3, 0.5, 0.7\}$, the hidden size from $\{64, 256, 512\}$, the heads of attention from $\{1, 2, 4\}$, M from $\{30, 50\}$, K from $\{5, 10\}$, rb_order from $\{1, 2\}$ and the number of layers from $\{1, 2, 3, 4, 5, 6, 7, 8, 9, 10\}$.

**SGFormer.**[8] We set the learning rate to 0.001, the number of global layer to 1 and $\alpha = 0.5$. We perform hyperparameter tuning on dropout rate from $\{0.1, 0.3, 0.5, 0.7\}$, the hidden size from $\{64, 256, 512\}$, the heads of attention from $\{1, 2, 4\}$, graph weight from $\{0.5, 0.8\}$, and the number of local layers from $\{1, 2, 3, 4, 5\}$.

**Polynormer.**[9] For Cora, Citeseer, Pubmed, Squirrel and Chameleon, we set the learning rate to 0.001. We perform hyperparameter tuning on dropout/indropout rate from $\{0.1, 0.3, 0.5, 0.7\}$, the hidden size from $\{64, 256, 512\}$, the heads of attention from $\{4, 6, 8\}$, weight decay from $\{0.005, 0.0005, 0.00005\}$, and the number of local/global layers from $\{1, 2, 3, 4, 5, 6, 7, 8, 9, 10\}$. For the remaining datasets, we use the officially provided hyperparameters without further modification.

**Spexphormer.**[10] For Cora, Citeseer, Pubmed, Wikics, Squirrel, Chameleon, Amazon-Ratings, Roman-Empire, Questions and ogbn-products, we perform hyperparameter tuning on the learning rate $\{0.001, 0.005\}$, the number of GNN/GT layers from $\{1, 2, 3, 4, 5\}$, dropout rate from $\{0.1, 0.3, 0.5\}$, the hidden size from $\{64, 256, 512\}$, weight decay from $\{0.005, 0.0005, 0.00005\}$ and the heads of attention from $\{2, 4, 6\}$. For the remaining datasets, we also use the officially provided hyperparameters without further modification.

## C.3 Computing Environment

Our implementation is based on PyG [17] and DGL [58]. The experiments are conducted on Python 3.8, Intel(R) Core(TM) i7-6700 CPU 3.40GHz 3.41 GHz, 24GB and Ubuntu 20.04 with Intel(R) Xeon(R) Platinum 8480+ CPU and 8 NVIDIA H800 GPUs.

---

[2] https://github.com/LUOyk1999/tunedGNN

[3] https://github.com/rampasek/GraphGPS

[4] https://github.com/JHL-HUST/NAGphormer

[5] https://github.com/hamed1375/Exphormer

[6] https://github.com/devnkong/GOAT

[7] https://github.com/qitianwu/NodeFormer

[8] https://github.com/qitianwu/SGFormer

[9] https://github.com/cornell-zhang/Polynormer

[10] https://github.com/hamed1375/Sp_Exphormer

Table 12: Dataset-specific hyperparameter settings of G$^2$Former. "BN" and "LN" denote batch normalization and layer normalization, respectively.

| Dataset | ResNet | Normalization | Dropout | GNNs layer | Hidden size | lr | $T$ | GNN backbone |
|---------|--------|---------------|---------|------------|-------------|-----|-----|--------------|
| Cora | False | False | 0.9 | 3 | 512 | 0.001 | 4 | GCN |
| Citeseer | False | False | 0.5 | 2 | 512 | 0.001 | 3 | GCN |
| Pubmed | False | False | 0.7 | 2 | 512 | 0.005 | 3 | GCN |
| Computer | False | LN | 0.5 | 2 | 512 | 0.001 | 5 | GCN |
| Photo | True | LN | 0.5 | 5 | 128 | 0.001 | 2 | GAT |
| CS | True | LN | 0.4 | 2 | 256 | 0.001 | 3 | GAT |
| Physics | True | LN | 0.6 | 2 | 256 | 0.001 | 3 | GCN |
| Wikics | False | LN | 0.7 | 2 | 128 | 0.001 | 7 | GAT |
| Squirrel | True | BN | 0.7 | 4 | 512 | 0.01 | 2 | GCN |
| Chameleon | False | False | 0.7 | 2 | 512 | 0.005 | 2 | GAT |
| Roman-Empire | True | BN | 0.3 | 9 | 512 | 0.001 | 3 | GCN |
| Amazon-Ratings | True | BN | 0.5 | 3 | 512 | 0.001 | 1 | GAT |
| Minesweeper | True | BN | 0.1 | 17 | 64 | 0.01 | 4 | GAT |
| Questions | True | False | 0.3 | 13 | 512 | 3e-5 | 5 | GCN |
| ogbn-arxiv | True | BN | 0.5 | 6 | 512 | 0.0005 | 3 | GCN |
| ogbn-products | False | LN | 0.5 | 3 | 256 | 0.003 | 3 | GraphSAGE |
| Pokec | True | BN | 0.2 | 9 | 256 | 0.0005 | 3 | GCN |

# D   Hyperparameters and Reproducibility

Since our GNN backbone is based on the three classical architectures, G$^2$Former's hyperparameter settings primarily follow [36], with minor modifications depending on the specific dataset. We also follow prior works regarding the use of mini-batch or full-batch training: except for ogbn-products and pokec, all datasets are trained in full-batch mode. For G$^2$Former-specific hyperparameters, such as the number of channels $T = \alpha + \beta$, manual tuning is required. Detailed configurations for reproducibility are provided in Table 12.

# E   Limitations

This study is based on a node classification benchmark and focuses solely on node-level tasks, lacking exploration of link prediction or graph classification. Establishing edge-level and graph-level evaluation benchmarks and further investigating the potential of GTs in supporting such tasks would be a valuable direction for future research.

