# OpenReview forum: "Restricted Global-Aware Graph Filters Bridging GNNs and Transformer for Node Classification"
_NeurIPS.cc/2025/Conference — NeurIPS 2025 poster_

### Official Review · Reviewer_gika · 2025-06-30

**Clarity:** 3
**Significance:** 2
**Originality:** 2
**Rating:** 5
**Confidence:** 5

**Summary:**

This paper proposes G2Former, a novel framework that integrates graph spectral filtering with Transformers to improve node classification performance. The method introduces multi-channel band-pass filters to enhance feature representation and reduce topological noise in attention mechanisms. Extensive experiments on 17 diverse datasets demonstrate the effectiveness and generalization ability of the approach.

**Questions:**

see in the weaknesses

**Ethical Concerns:**

["NO or VERY MINOR ethics concerns only"]

**Final Justification:**

It has addressed the issues I raised. There are minor flaws in the writing.

**Limitations:**

yes

**Quality:**

2

**Strengths And Weaknesses:**

Strengths:

The paper conducts experiments on 17 datasets, including homophilous, heterophilous, and large-scale graphs, which demonstrates the broad applicability and robustness of the proposed method.

By introducing multi-channel graph filters with band-pass properties, the method effectively reduces topological noise in attention and enhances the generalization ability of GNNs.

Weaknesses：

Line 37：

There is an obvious grammatical error: the phrase "have have" is duplicated.

Line 8：

The claim that "the global attention module does not provide significant performance gains" is overly strong and lacks sufficient theoretical or empirical support. Many existing works (e.g., Graphormer, GTN) have shown the effectiveness of global attention in complex graph structures. Moreover, the proposed method itself incorporates global attention, which contradicts the claim.

Line 12：

The paper asserts that "the Transformer improves generalization in GNNs" as a core contribution, but no experiments are designed to directly measure generalization. Therefore, the claim is unsubstantiated.

Line 130：

The symmetric attention matrix
A_s is not truncated or sparsified.If all attention values are non-zero and the graph is fully connected, does the degree matrix D_S result in all nodes having the same degree equal to the number of nodes?

Necessity of Attention Mechanism Unclear:

The proposed G2Former effectively implements a learnable multi-channel graph filter. This could potentially be achieved using parametric spectral filters directly. Whether the attention mechanism is necessary for constructing the adjacency matrix is not discussed. If attention is only used to derive a spectral filter, there might be more direct and efficient alternatives, which should be addressed.

---

> ### Author Rebuttal · Authors · 2025-07-30
>
> **Weakness 1.** Thank you for catching the typo. We will correct the duplicated phrase "have have" at line 37 in the final version.
>
> **Weakness 2.** We appreciate the reviewer’s concern, and we would like to clarify a potential misunderstanding.
> The statement that “global attention modules do not provide significant performance gains” is in fact well-supported by existing empirical evidence.
> First, as cited in the Introduction (Line 56), prior work **[1]** has shown that many Graph Transformer models fail to outperform carefully tuned GNNs across datasets with diverse properties — including homophily, heterophily, and large-scale graphs.
> Second, our own empirical study (Section 5), which includes comprehensive benchmarks across 17 datasets as well as training and test loss curves, further reinforces this observation.
> Additionally, while Graphormer(NeurIPS 2021) and GTN(NeurIPS 2019) were not included in the original version due to their earlier publication dates, we have now added their results for comparison.
> The experimental results are shown in the table below, where the performance of Graphormer and GTN remains inferior to that of the fine-tuned GNNs and our proposed method.
> **Most importantly, we would like to reiterate the core goal of our work:
> To investigate whether the Transformer architecture — and specifically the global attention module — provides substantial performance improvements in GNNs. This is a problem that the GNN community must face and respond to.**
> Our key finding is nuanced:
> **Global attention alone does not reliably improve GNN performance. However, when constrained through a frequency-aware design, it can significantly support generalization.**
> This challenges prevailing assumptions in the community, and highlights the need to revisit the role of attention mechanisms in GNNs.
> **Therefore, we emphasize that our inclusion of a global attention module is intentional and essential for our analysis, not contradictory.**
> It allows us to reinterpret attention as a spectral operator and to examine when and why it may help GNNs generalize — under controlled conditions.
>
> **Table: Performance comparison with Graphormer and GTN.**
> The best performance is **bolded**, and the suboptimal performance is marked in *italics*. The tuned GNN uses the best-performing classic GNN, as indicated in parentheses.
>
> | **Model**     | **Cora**                    | **Citeseer**               | **Photo**                        | **Squirrel**                  | **Chameleon**                 |
> |---------------|-----------------------------|----------------------------|----------------------------------|-------------------------------|-------------------------------|
> | **Metric**    | Accuracy                    | Accuracy                   | Accuracy                         | Accuracy                      | Accuracy                      |
> | **Type**      | Homophilous                | Homophilous               | Homophilous                     | Heterophilous                | Heterophilous                |
> | **GTN**        | 80.36${\pm}$0.9            | 68.76${\pm}$1.5           | 89.93${\pm}$0.6                 | 38.42${\pm}$2.6             | 36.37${\pm}$3.1             |
> | **Graphormer** | 83.15${\pm}$0.7            | 72.11${\pm}$0.8           | 92.70${\pm}$0.4                 | 35.33${\pm}$2.3             | 37.81${\pm}$3.6             |
> | **tuned GNN**  | **84.56**${\pm}$0.7(GCN)   | 72.28${\pm}$0.5(GCN)      | 96.71${\pm}$0.30(GraphSAGE)  | 45.03${\pm}$1.7(GCN)        | **44.91**${\pm}$4.6(GCN)    |
> | **Ours**       | *83.72* ${\pm}$0.6     | **72.62**${\pm}$0.6       | **97.06**${\pm}$0.1             | **45.53**${\pm}$1.8         | *44.34* ${\pm}$4.4       |
>
>
> **Weakness 3.** Thank you for pointing out this important issue.
> We believe there may be a misunderstanding regarding how we evaluate generalization. While we understand the concern, we interpret generalization in an empirical context, as is standard in GNN evaluation. The current challenges faced by Graph Transformer are based on an empirical research[1], so this approach will be a direct and powerful response.
> In fact, we explicitly analyze generalization behavior in Section 4 through the following means:
>
> **The gap between test losses of different models.**
>
> **The stability of test-time performance across datasets with varying graph structures (homophily, heterophily, scale).**
>
> **Visualization of the attention matrix under multi-channels constraints.**
>
> These evaluation strategies are consistent with prior empirical studies in the community. **In response to your concerns, we further provide the generalization gap **[2]** of the model as shown below.** Our method consistently achieves the lowest generalization gap. Therefore,  we believe our current experiments sufficiently and appropriately capture the generalization behavior of both GNNs and Transformer-based models in practice.
>
> **Table: Generalization gap on homophilous graphs.**
>
> | **Model**     | **Computer** | **Photo** | **CS** | **Physics** | **WikiCS** |
> |---------------|--------------|-----------|--------|-------------|------------|
> | **SGFormer**  | 0.2650       | 0.1356    | 0.0963 | 0.0719      | 0.9439     |
> | **Polynormer**| 0.1454       | 0.1622    | 0.0574 | 0.0911      | 0.8236     |
> | **tuned GNN** | 0.0893 (GAT) | 0.2341 (GraphSAGE) | 0.0874 (GraphSAGE) | 0.0089 (GCN) | 0.4402 (GAT) |
> | **G$^2$Former (Ours)** | 0.0686$\downarrow$ | 0.0053$\downarrow$ | 0.0196$\downarrow$ | 0.0051$\downarrow$ | 0.3657$\downarrow$ |
>
> **Table: Generalization gap on heterophilous graphs.**
>
> | **Model**      | **amazon\_ratings** | **roman\_empire** | **minesweeper** | **questions**      |
> |----------------|---------------------|-------------------|-----------------|--------------------|
> | **SGFormer**   | 8.2326              | 0.8225            | 0.3561          | 1.7627             |
> | **Polynormer** | 3.3355              | 0.6561            | 0.3393          | 0.7404             |
> | **tuned GNN**  | 4.0764 (GAT)        | 0.6327 (GCN)      | 0.3475 (GCN)    | 0.3036 (GCN)       |
> | **G$^2$Former (Ours)** | 2.0837$\downarrow$  | 0.4242$\downarrow$| 0.0900$\downarrow$ | 0.2411$\downarrow$ |
>
> **Weakness 4.** We thank the reviewer for this insightful observation, which touches on a fundamental challenge in the design of Graph Transformers.
> Indeed, $\mathbf{A}\_s$ is not explicitly truncated or sparsified in our implementation — and this is by design.
> First, although $\mathbf{A}\_s$ is dense, the attention weights are derived from softmax-normalized scores, which typically concentrate on a few dominant entries. This results in a non-uniform degree matrix $\mathbf{D}\_s$ and a meaningful spectrum for $\mathbf{L}\_s$.
> Enforcing sparsity would require explicit construction and thresholding of $\mathbf{A}\_s$, incurring $\mathcal{O}(n^2)$ complexity — which we intentionally avoid for scalability.
>
> Second, we appreciate the reviewer’s sharp insight, **which touches on a subtle but critical issue also raised in our Introduction — namely, the “representation laziness” of dense, unconstrained attention.**
> Prior models such as SGFormer and Polynormer allow the attention module to freely learn graph inductive biases, but in practice this often leads to overly smooth or structure-agnostic behavior.
> Our work directly addresses this challenge by reinterpreting the attention matrix as a spectral operator, and constraining its response to enhance generalization of GNN rather than relying on unstructured learning alone.
>
> In this light, we view $\mathbf{A}\_s$ as a frequency modulation tool — and preserving its dense form is essential for analyzing its behavior.
>
> **Weakness 5.** We sincerely thank the reviewer for raising this thoughtful question.
> We believe this point touches on a deeper misunderstanding about the goal of our paper.
>
> This work is not primarily intended to introduce a novel method that achieves state-of-the-art performance, but rather to address a critical open question in the community:
>
> **Can Transformer architectures truly help GNNs break through existing performance ceilings?** This was challenged by an empirical study **[1]**.
>
> To answer this, we design G2Former as a controlled analytical framework, where we reinterpret the attention matrix not as a learned graph structure, but as a spectral filter — and systematically study its effect under frequency constraints.
>
> **We emphasize that this design is not about replacing the adjacency matrix with a better one, but about exposing and understanding the actual role of global attention in generalization of GNN, which has often been regarded as having broken through the performance bottleneck of GNNs in prior works.**
>
> As we note in the Introduction, many existing Graph Transformers rely on unconstrained attention modules to "implicitly learn" the topology.
> However, this often results in representation laziness, as also observed in SGFormer and Polynormer.
> Our work critically examines this phenomenon and shows that without proper spectral constraints, attention alone provides limited benefit.
>
> **Therefore, we intentionally retain the attention module — not because it is the most efficient or direct way to build a graph filter, but because it is the object of our study.**
> We hope this clarifies that our paper is driven by scientific inquiry into an open and impactful question, rather than by methodological novelty alone. Thank you again for your comment and we sincerely hope you can refer to our reply.
>
> [1] Classic gnns are strong baselines: Reassessing gnns for node classification. NeurIPS 2024.
>
> [2] Graph neural networks are inherently good generalizers: Insights by bridging gnns and mlps. ICLR 2023.

---

> > ### Comment · Reviewer_gika · 2025-08-06
> >
> > Thank you for your detailed responses to my review. I appreciate the clarifications and additional results you provided. However, I still have several concerns.
> >
> > w2:
> >
> > As you mentioned in your response, the statements "global attention modules do not provide significant performance gains" and "Global attention alone does not reliably improve GNN performance" appear to be contradictory. So, do you believe that global attention can improve performance or not? The experimental results in the tables (Ours vs. GNNs) seem to support both claims, which adds to the confusion.
> >
> > w4:
> >
> > a. If A_s is a dense matrix, then the graph is fully connected, and D_s would be a diagonal matrix with identical diagonal entries. Why would D_s then be non-uniform?
> >
> > b. Why does enforcing sparsity by explicitly constructing and thresholding A_s increase complexity? It seems that simply introducing a threshold hyperparameter or a learnable parameter could zero out some edge weights and achieve sparsity. Moreover, in the paper, A_s itself is already fully connected, correct? Additionally, are the model and util files in the anonymous link too large? I am unable to load them.
> >
> > c. In SGFormer, the transformer component and the GNN component are separate. Both parts take the raw input features, and their outputs are simply concatenated or added before the final classification layer. This structure does not fall into either of the two categories mentioned in lines 55–60 of the paper. This can be verified in the official SGFormer code. Consequently, this undermines the logical consistency of the subsequent analysis in the paper.
> >
> > w5:
> >
> > Thank you for your reply. There might have been a small misunderstanding. What I am curious about is this: while the authors argue that spectral filters can benefit the global attention mechanism, could it be that the global attention mechanism, in turn, limits the effectiveness of the spectral filters themselves? It would be worth discussing whether directly applying parameterized spectral filters could achieve better performance.

---

> ### Author Response · Authors · 2025-08-06
>
> Dear Reviewer,
>
> Thank you again for your time and effort in reviewing our paper.
>
> We’ve submitted our rebuttal addressing the comments provided, and we’d greatly appreciate it if you could take a moment to consider our responses.
>
> Please feel free to let us know if you have any remaining questions or need clarification on any point. Thank you once again for your valuable feedback.

---

> ### Author Response · Authors · 2025-08-07
>
> We sincerely thank you for your continued attention and thoughtful responses to our work!
>
> **w2.** We would like to clarify our perspective. Both statements aim to convey that, despite incorporating global attention mechanisms, graph Transformers still struggle to surpass the empirical performance ceiling of tuned traditional GNNs. However, when appropriate constraints are imposed, such mechanisms can contribute to improving generalization. Therefore, we argue that global attention modules may not empirically boost the performance upper bound of GNNs, but they can serve as regularization components to enhance generalization and mitigate overfitting on GNNs. We respectfully hope that our explanation has clarified our intentions.
>
> **w4.**
>
> **a.** Although the attention matrix $\mathbf{A}_{S}$ is dense in this case, the inner product between a node's representation and those of other nodes varies, leading to non-uniform entries in $\mathbf{A}_S$. As a result, the row sums of $\mathbf{A}\_S$ are not equal. In Figure 5 of our paper, we visualized the attention matrix of the model on the Minesweeper dataset. **Taking this graph as an example, we further report the maximum (9389.3), minimum (8436.0), mean (9094.6), and standard deviation (320.4) of the diagonal entries of its corresponding $\mathbf{D}\_S$.** We hope this can address your concern.
>
> **b.** This issue was explained in Equation (1) of our paper. **Leveraging the associativity of matrix multiplication, we rearrange the computation order to avoid explicitly forming the attention matrix. Although the attention matrix is theoretically dense, it is never explicitly computed; all operations are performed implicitly. It has also been adopted in SGFormer and Polynormer.** As we clarify in lines 223–230, if the attention matrix $\mathbf{A}_{S}$ were explicitly constructed, both the time and space complexity would be $O(\left | \mathcal{V}  \right | ^2)$. Thus, the key lies not in setting a threshold or introducing a learnable parameter, but in the design of the computation itself. Regarding the anonymous GitHub repository, the access delay might be caused by server load. To further address your concerns, we provide the relevant portion of the code below for your direct inspection:
> ```python
>     D_invsqrt = torch.pow(
>         torch.mm(q, torch.mm(q.t(), torch.ones(feat.shape[0], 1, device=feat.device))) - 1 + 1e-10,
>         -0.5
>     )
>
>     h = torch.mm(q.t(), feat * D_invsqrt)
>     h = torch.mm(q, h)
>     h = D_invsqrt * h
> ```
> To facilitate the explanation, let us assume $\mathbf{Q}'=\mathbf{K}'$, and denote $\mathbf{Q}'$ as `q`, $f\_{\theta\ _2}(\mathbf{X})$ as `feat`, and $\mathbf{D}\_S^{{-1/2}}$ as `D_invsqrt`. The above code corresponds to the following equation:
>
> $\mathbf{D}\_S^{-1/2}\mathbf{A}\_S\mathbf{D}\_S^{-1/2}f\_{\theta\ _2}(\mathbf{X})=\mathbf{D}\_S^{-1/2}(\mathbf{Q'}(\mathbf{K'}^T(\mathbf{D}\_S^{-1/2}f\_{\theta\ _2}(\mathbf{X}))))$.
>
> This reordering of the computation only requires a complexity of $O(\left | \mathcal{V}  \right |)$.
>
> **c.** As you mentioned, the attention component and the GNN component in SGFormer are separate modules, each encoding independently before being fused. Since there is no sequential or hierarchical dependency between them, the attention-encoded output effectively serves as a **supplement** to the GNN component. **Following Polynormer’s classification scheme, SGFormer fits the local-and-global paradigm, where “and” signifies the independent roles of the two components.** In contrast, Polynormer adopts a local-to-global approach, first encoding with the GNN module and then feeding the result into the attention module for secondary encoding, where “to” indicates a sequential and dependent relationship between the two. We have also illustrated this distinction in Figure 1 of our paper, thus no logical inconsistency exists.
>
> **w5.** Thank you very much for your understanding! First, we would like to clarify that our work reinterprets the attention matrix as a global graph filter with the purpose of imposing spectral constraints (i.e., multi-channel band-pass filtering) on it. Therefore, we do not initialize new filters based on the original graph structure, and the concern of “global attention mechanism, in turn, limits the effectiveness of the spectral filters themselves?” does not apply in our context.
>
> That said, we appreciate your suggestion that studying the role of parameterized graph filters in this scenario could be an interesting direction—though it falls outside the scope of our current work, which primarily focuses on the practical role of Transformers in GNNs. For performance comparisons with certain spectral GNNs, please refer to our response to Reviewer g3CZ, Q4.
>
> We hope that the intermediate outputs, code inspection, and related analyses provided above can help alleviate your concerns. We sincerely appreciate your thoughtful engagement and close attention to our work.

---

> ### Comment · Reviewer_gika · 2025-08-07
>
> Thank you very much for your reply. It has addressed most of my concerns. I suggest refining certain details in the final version to avoid potential misunderstandings. I have adjusted my rating to 5.

---

> > ### Author Response · Authors · 2025-08-08
> >
> > Thank you very much for your thoughtful feedback and the adjusted rating. We will carefully refine the details in the final version to ensure clarity and avoid potential misunderstandings. Your comments are greatly appreciated.

---

### Official Review · Reviewer_g3CZ · 2025-07-02

**Clarity:** 2
**Significance:** 3
**Originality:** 3
**Rating:** 4
**Confidence:** 3

**Summary:**

This paper explores the global attention mechanism in Graph Transformers in terms of node representation. The authors identify the test error oscillation phenomenon led by the global attention component and point out that mitigating such oscillation improves the generalization in GNNs. Those insights are evaluated in comprehensive experimental evaluations.

**Questions:**

1.	The rationale of global-to-local design is not fully elaborated. In the spectral design, G2Former utilizes the global attention matrix in place of the adjacency matrix. I understand that the filtered signals are fed into GNNs as inputs. However, the original graph structure is not considered at this spectral filtering stage. The authors should justify the rationale for this design.

2.	The theoretical analysis or conclusion on the generalization in GNNs is missing. The ability of the Transformer to improve the generalization in GNNs claimed in the paper, is critical. I suggest the authors provide the necessary theoretical analysis.

3.  It lacks necessary explanations of G2Former’s inferior performance on homophily graphs compared with other graph-transformer-based models. G2Former does not perform well on the three homophily datasets. Please justify this phenomenon.

4.	Some relevant spectral GNN models are not discussed or compared. Spectral GNNs acting as spectral graph filters on graph signals are closely relevant to the scope of this paper. However, they are not mentioned or compared in the experiments. For example, for example [1][2][3][4].

[1] BernNet: Learning arbitrary graph spectral filters via Bernstein approximation. NeurIPS 2021

[2] How powerful are spectral graph neural networks? ICML 2022

[3] Graph neural networks with learnable and optimal polynomial bases, ICML, 2023

[4] How Universal Polynomial Bases Enhance Spectral Graph Neural Networks: Heterophily, Over-smoothing, and Over-squashing. ICML 2024

**Ethical Concerns:**

["NO or VERY MINOR ethics concerns only"]

**Final Justification:**

My major concerns have been cleared.

**Limitations:**

Yes

**Quality:**

2

**Strengths And Weaknesses:**

Strength:

1. This paper proposed an interesting idea of incorporating global attention in the Transformer into the design of spectral GNN.

2. Theoretical analyses on the spectral properties of G2Former and its time complexity are provided, which provide some insights into the research of graph transformers.

3. Extensive evaluation of G2Former is conducted against both graph transformer baselines and GNNs on homophily and heterophily datasets.


Weakness:

1. The rationale of global-to-local design is not fully elaborated.

2. The theoretical analysis or conclusion on the generalization in GNNs is missing.

3. It lacks necessary explanations of G2Former’s inferior performance on homophily graphs compared with other graph-transformer-based models.

4. Some relevant spectral GNN models are not discussed or compared.

---

> ### Author Rebuttal · Authors · 2025-07-30
>
> **Q1.** We thank the reviewer for the insightful question regarding the use of the global attention matrix in place of the original adjacency matrix during the spectral filtering stage. First, **we would like to clarify that this design follows a well-established paradigm in Graph Transformers, where the attention matrix is commonly used as the graph structure within attention-based message passing or implicit filtering modules.**
> Our contribution lies not in the replacement itself, but in providing a novel spectral interpretation of this process:
> We reinterpret the attention matrix as a learnable graph filter, allowing us to impose explicit frequency constraints (e.g., multi-band-pass filtering), and use it to inject generalization-enhancing signals into the node features prior to GNN layers.
> The learned global matrix replaces the adjacency only within the auxiliary filtering step. The original graph structure is fully preserved in downstream message-passing, so our method can still leverage its topological priors.
> This decoupling enables the model to build a complementary structural view that enhances generalization, especially under structural noise or heterophily. We will clarify this distinction and its motivation in the revised version to prevent confusion.
>
> **Q2.** We appreciate the reviewer’s suggestion, and agree that theoretical analysis of generalization remains an important and open challenge in the GNN community.
> At the same time, we would like to highlight a long-standing gap between theoretical predictions and empirical performance in this field.
> For instance, **[1] provides theoretical justification that self-attention improves the generalization ability of Graph Transformers over GCNs. However, this conclusion is not always supported in practice. This is due to the fact that this generalization analyses focus on a narrow class of simplified Transformer architectures, without considering key components used in actual models (e.g., common tricks such as residual connections, layernorm/batchnorm, dropout, etc.). These components have been demonstrated in **[3,5]** to play a critical role in unlocking the full potential of GNNs.**
> Given this gap, our work focuses on practical indicators of generalization, such as:
> Generalization across diverse datasets (Tables 1–3);
> Stability in test loss dynamics (Figure 4);
> Visualization of the Attention Matrix under Regularization Constraints (Figure 5).
> Additionally, **to alleviate your concerns, we report the model's generalization gap **[4]** in the table below. Our method consistently achieves the lowest generalization gap.**
> Rather than proposing abstract theoretical bounds, we provide empirical and spectral perspectives on how Transformer modules interact with graph inductive biases — offering a mechanism-level understanding that complements existing theoretical views.
> We respectfully invite the community to revisit the reliability and applicability of current theoretical frameworks, with the goal of more effectively advancing modern GNN research.
>
> **Table: Generalization gap on homophilous graphs.**
>
> | **Model**     | **Computer** | **Photo** | **CS** | **Physics** | **WikiCS** |
> |---------------|--------------|-----------|--------|-------------|------------|
> | **SGFormer**  | 0.2650       | 0.1356    | 0.0963 | 0.0719      | 0.9439     |
> | **Polynormer**| 0.1454       | 0.1622    | 0.0574 | 0.0911      | 0.8236     |
> | **tuned GNN** | 0.0893 (GAT) | 0.2341 (GraphSAGE) | 0.0874 (GraphSAGE) | 0.0089 (GCN) | 0.4402 (GAT) |
> | **G$^2$Former (Ours)** | 0.0686$\downarrow$ | 0.0053$\downarrow$ | 0.0196$\downarrow$ | 0.0051$\downarrow$ | 0.3657$\downarrow$ |
>
> **Table: Generalization gap on heterophilous graphs.**
>
> | **Model**      | **amazon\_ratings** | **roman\_empire** | **minesweeper** | **questions**      |
> |----------------|---------------------|-------------------|-----------------|--------------------|
> | **SGFormer**   | 8.2326              | 0.8225            | 0.3561          | 1.7627             |
> | **Polynormer** | 3.3355              | 0.6561            | 0.3393          | 0.7404             |
> | **tuned GNN**  | 4.0764 (GAT)        | 0.6327 (GCN)      | 0.3475 (GCN)    | 0.3036 (GCN)       |
> | **G$^2$Former (Ours)** | 2.0837$\downarrow$  | 0.4242$\downarrow$| 0.0900$\downarrow$ | 0.2411$\downarrow$ |
>
> **Q3.** We appreciate the reviewer's question. Indeed, G2Former does not outperform all baselines on the three old toy homophily-dominated datasets (Cora, Citeseer, PubMed), and we think this is an expected and informative result, rather than a shortcoming.
> These datasets are known for their high homophily, which align closely with the inductive bias of classical GNNs — and also favor Transformer models that preserve this local bias.
> Our architecture includes graph-guided filtering (low and high frequency component) — which may not provide additional benefit (and could introduce unnecessary flexibility) on highly smooth graphs.
> Additionally, we intentionally avoid overfitting to citation benchmarks through extensive per-dataset hyperparameter tuning, which has been criticized in **[2]** as a source of misleading claims. This is the case for most GTs (except for Polynormer, whose sourse paper avoided referring to these three homophilous datasets). In our response to R1of Reviewer ieX9, we presented the homophily rankings of these three graphs within homophilous datasets, where they ranked among the top three.
> We will clarify this observation in the final version and emphasize that G2Former is not optimized for a single data regime, but rather for broader generalization across graph types. We believe this reflects an important tradeoff in model design: performance consistency vs task-specific overfitting.
>
> **Q4.** We thank the reviewer for pointing out the connection to spectral GNNs, such as BernNet, JacobiConv, OptBasisGNN, UniFilter. While our method is indeed inspired by graph signal processing, we would like to clarify that our setting and objectives differ significantly from conventional spectral GNNs.
> Classical spectral GNNs typically rely on static graph filters derived from the input Laplacian or pre-defined polynomial bases, and assume a relatively stable global frequency profile.
> In contrast, G2Former uses a learned attention-based Laplacian to construct adaptive, input-dependent graph filters, enabling more flexible and localized frequency manipulation.
> Furthermore, our work focuses on providing a spectral interpretation of the attention mechanism in graph Transformers, aiming to address the effectiveness dilemma currently faced by such models. We also provide a comparison between our method and these spectral GNNs, as shown in the table below. The best performance is highlighted in bold. Our method still demonstrates superior performance. We hope these will address your concerns.
>
> **Table: Comparison of model performance on homophilous graphs.**
>
> | **Model**      | **Computer**         | **Photo**            | **cs**               | **physics**          | **wikics**           |
> |----------------|----------------------|-----------------------|-----------------------|-----------------------|-----------------------|
> | **BernNet**     | 92.63$_{\pm 0.5}$     | 94.65$_{\pm 0.2}$      | 94.77$_{\pm 0.3}$      | 96.54$_{\pm 0.1}$      | 75.29$_{\pm 0.5}$      |
> | **JacobiConv**  | 92.32$_{\pm 0.2}$     | 93.76$_{\pm 0.1}$      | 94.21$_{\pm 0.5}$      | 96.17$_{\pm 0.1}$      | 75.47$_{\pm 1.1}$      |
> | **OptBasisGNN** | 91.04$_{\pm 0.4}$     | 95.31$_{\pm 0.3}$      | 95.66$_{\pm 0.6}$      | 96.81$_{\pm 0.1}$      | 77.63$_{\pm 0.9}$      |
> | **UniFilter**   | 93.20$_{\pm 0.3}$     | 93.98$_{\pm 0.4}$      | 93.22$_{\pm 0.1}$      | 96.77$_{\pm 0.3}$      | 78.82$_{\pm 0.7}$      |
> | **G$^2$Former** | **94.29**$_{\pm 0.1}$ | **97.06**$_{\pm 0.1}$  | **96.53**$_{\pm 0.1}$  | **97.60**$_{\pm 0.1}$  | **81.14**$_{\pm 0.2}$  |
>
> **Table: Comparison of model performance on heterophilous graphs.**
>
> | **Model**      | **amazon_ratings**      | **roman_empire**       | **minesweeper**         | **questions**           |
> |----------------|--------------------------|--------------------------|--------------------------|--------------------------|
> | **BernNet**     | 50.70$_{\pm 0.6}$         | 88.71$_{\pm 0.7}$         | 89.76$_{\pm 0.5}$         | 76.04$_{\pm 0.7}$         |
> | **JacobiConv**  | 52.39$_{\pm 0.4}$         | 89.32$_{\pm 0.3}$         | 91.62$_{\pm 0.3}$         | 76.11$_{\pm 1.0}$         |
> | **OptBasisGNN** | 52.11$_{\pm 0.4}$         | 88.94$_{\pm 0.6}$         | 93.50$_{\pm 1.2}$         | 77.23$_{\pm 0.8}$         |
> | **UniFilter**   | 53.66$_{\pm 0.7}$         | 91.32$_{\pm 0.6}$         | 92.38$_{\pm 0.7}$         | 77.56$_{\pm 0.9}$         |
> | **G$^2$Former** | **55.86**$_{\pm 0.1}$     | **93.03**$_{\pm 0.3}$     | **99.45**$_{\pm 0.1}$     | **79.55**$_{\pm 0.3}$     |
>
> [1] What Improves the Generalization of Graph Transformers? A Theoretical Dive into the Self-attention and Positional Encoding. ICML 2024.
>
> [2] Position: Graph Learning Will Lose Relevance Due To Poor Benchmarks. ICML 2025.
>
> [3] Can Classic GNNs Be Strong Baselines for Graph-level Tasks? Simple Architectures Meet Excellence. ICML 2025.
>
> [4] Graph neural networks are inherently good generalizers: Insights by bridging gnns and mlps. ICLR 2023.
>
> [5]  Classic gnns are strong baselines: Reassessing gnns for node classification. NeurIPS 2024.

---

> > ### Comment · Reviewer_g3CZ · 2025-08-06
> > **Ack**
> >
> > I appreciate the detailed responses from the authors. My concerns are partially addressed. However, I still have a few concerns. First, you regard the "attention matrix as a learnable graph filter", which indicates the model can learn a suitable graph filter according to the spectral property of underlying graphs. You later claim that "may not provide additional benefit on highly smooth graphs". These two statements seem to contradict each other.
> > Second, in your responses to Q4, you state "Classical spectral GNNs typically rely on static graph filters derived from the input Laplacian or pre-defined polynomial bases". As far as I know, this is not the case for OptBasisGNN and UniFilter.
> > Third, since OptBasisGNN is also a learnable graph filter, I am curious about the significant advantage of your proposed model as a learnable graph filter by utilizing the attention matrix over OptBasisGNN.
> > Thanks.

---

> ### Author Response · Authors · 2025-08-06
>
> Dear Reviewer,
>
> I hope this message finds you well.
>
> We would like to kindly remind you that we have submitted our rebuttal for our paper. We sincerely appreciate the time and effort you have dedicated to reviewing our work, and we hope that our responses address your concerns and clarify any questions you raised.
>
> Should you have any additional questions or concerns, we would be happy to provide further clarification.
>
> Thank you again for your valuable feedback and contribution to the review process.

---

> ### Author Response · Authors · 2025-08-06
>
> Thank you again for your insightful follow-up. We would like to further clarify the key differences between our proposed method and existing spectral GNNs, particularly OptBasisGNN, as well as the intended meaning behind our use of the term "learnable graph filter."
>
> 1. **Our use of this term refers specifically to the attention matrix learned by the Transformer (driven by node features), not to any filter derived from the eigenvalues or eigenvectors of the original graph operator (as is the case in spectral GNNs).** Importantly, our method is completely decoupled from the original graph topology during the filtering process. Instead, **we treat the attention matrix as a latent, learnable structure**, and we apply multi-bandpass spectral constraints to regularize its frequency characteristics. This is designed to mitigate the effect of spurious or noisy connections in the attention matrix, which can otherwise degrade the performance of downstream GNNs.
>
> We agree that the original phrasing was unclear. What we intended to convey is that when the input graph is highly smooth (i.e., has high homophily), introducing high-frequency components via multi-bandpass constraints may offer limited additional benefit, since a simple low-pass filter might already be sufficient. **This is not a contradiction, but a context-specific observation**, and we will revise the wording to avoid confusion.
>
> 2. When we mentioned that classical spectral GNNs rely on “static graph filters,” we meant that the filters are learned in the spectral domain of the original graph. In contrast, **our attention matrix is learned independently of the original graph topology (This form of decoupling represents a common paradigm in graph Transformers, e.g. SGFormer and Polynormer.),** and we apply spectral regularization before passing it into the GNN. Therefore, the “filter” in our model is not tied to the original graph spectrum, which makes it fundamentally different from prior approaches.
>
> 3. **Rather than proposing a better spectral filter per se, our work investigates the question:**
>
> **“Does the Transformer truly enhance the performance of GNNs?”**
>
> **This issue has been challenged by an empirical study [1].** An unconstrained attention matrix may capture noisy or spurious edges. Reinterpreting it as a graph filter provides a principled way to impose spectral constraints (i.e. multi-channel band  pass filtering). **Our goal is orthogonal to that of spectral GNNs like OptBasisGNN, which aim to design optimal filters based on graph topology.**
>
> We hope this clarifies the conceptual and technical distinctions between our method and prior spectral approaches, and we truly appreciate your thoughtful feedback.
>
> [1] Classic gnns are strong baselines: Reassessing gnns for node classification. NeurIPS 2024.

---

> > ### Comment · Reviewer_g3CZ · 2025-08-06
> >
> > Thank the authors for the replies. I have adjusted the score accordingly.

---

> > > ### Author Response · Authors · 2025-08-08
> > >
> > > We sincerely appreciate your thoughtful reconsideration and are grateful for the adjusted evaluation. Thank you again for your valuable feedback.

---

### Official Review · Reviewer_3Sa7 · 2025-07-03

**Clarity:** 2
**Significance:** 3
**Originality:** 3
**Rating:** 3
**Confidence:** 5

**Summary:**

The paper examines the efficacy of Graph Transformers (GT) in graph learning and proposes a new framework called G2Former, which redesign GT as band-pass global-aware graph filters. From the emprical study, the author shows that GT’s global attention does not significantly improve node classification performance and may introduce topological noise. Instead of using GT in traditional parallel or stacking fashions with GNNs, G2Former places the global attention module upfront, using it to generate band-pass filtered guidance signals which are then injected into downstream GNNs via a graph-guided filtering mechanism. Extensive experiments on 17 graph datasets (homophilous, heterophilous, large-scale) show improved performance.

**Questions:**

I have listed a few questions in the weakness part.

**Ethical Concerns:**

["NO or VERY MINOR ethics concerns only"]

**Final Justification:**

After considering all aspects of the paper, including the paper's clarity, method's adaptability across homophily, I’ve decided to keep my score unchanged.

**Limitations:**

yes.

**Paper Formatting Concerns:**

I noticed in the checklist, the author answer all the questions but didn’t provide any justification to each question.

**Quality:**

3

**Strengths And Weaknesses:**

Strengths:
1. The paper provided clear motivation and insightful empirical observations, like the (e.g., loss plots and visualizations) that GT often fail to significantly improve GNNs and may even degrade performance.
2. The paper introduce a theoretically grounded G2Former, which treats GT as band-pass graph filters. The mathematical formulation of band-pass filters with learnable graph spectra is well done and connects the model to spectral theory.
3. Extensive experiments span homophilous, heterophilous, and large graphs, and compare against a wide range of classical GNNs and SOTA graph Transformers.

Weakness:
1. Clarity of the methods can be improved. The method section is dense, with heavy reliance on mathematical notation; Some design choices suddenly appear without intuitive justification. For instance:
- the use of two FFNs (fθ1 and fθ2) is introduced abruptly without justification. What is the reason for using FFN instead of simple projection matrix? Will these increased parameters and enhanced representation power works as confounding factor and increase the performance?
- The graph-guided filtering mechanism is mathematically sound but lacks a clear conceptual description. Why this approach helps, what type of features it enhances, and how it interacts with heterophily? Some figures showing how the spectral response of your filter under different heterophily might help justify its usage.
- The paper can benefit from clearer section transitions, perhaps starting each technical section with a “motivation paragraph” that ties back to observed problems, then leads to your design choices..

2. The paper introduces three paradigms: local-to-global, local-and-global, and global-to-local. But the conceptual framing is underdeveloped, and not well described. Specifically, the authors do not clearly define or survey prior work falling into the first two categories, but only provide the framing in Figure 2, and leave the reader to figure it out. For example, SGFormer and Polynormer are mentioned as representatives, but there’s no summary of their architectural roles or design philosophies. A taxonomy of existing GT architectures with citations would make the novelty of G2Former’s global-to-local scheme more understandable and grounded. In its current form, the distinction feels more like a post-hoc categorization rather than a well-established analytical framework.

3. I notice that some newer or more tuned heterophily-centered GNN baselines (e.g., H2GCN, FAGCN) that perform well on heterophilous graphs were not included in the tables. Besides, the G2Former results are strong but not always dominant over GNNs on homophilous datasets, limiting claims of universal superiority.

---

> ### Author Rebuttal · Authors · 2025-07-30
>
> **W1.** We thank the reviewer for this important comment and the opportunity to clarify our design choices. The use of two FFNs follows the standard design in Transformer architectures **[1]**, where each attention layer is followed by a FFN. In our case, since the model is not designed as a stacked encoder, we directly apply FFNs to the node features. $f_{\theta _1}$ maps node features into the latent space for constructing the attention matrix; $f\_{\theta\_2}$ transforms features before graph spectral filtering, which is a standard preprocessing step for graph signal processing **[2][3]** and also adopted in some graph Transformers (e.g. SGFormer). **In fact, this common setting is rarely mentioned in previous papers. It is clarified in this paper for the sake of detail and reproducibility.**
> This separation allows us to independently control topology construction and feature adaptation under the filtering view. We provide a comparison of parameter size between our method and the best-performing graph Transformer baseline, Polynormer, as shown in two tables below. **Across all datasets, our approach consistently requires fewer parameters than Polynormer.** This efficiency primarily stems from the design where all spectral channels share the same pair of FFNs. Therefore, model complexity is well controlled. We sincerely thank the reviewer for the careful and responsible concern regarding this aspect.
> Hope these could address your concerns.
>
> **Table: Comparison of model parameter scales on homophilous graphs.**
>
> |  | **Computer** | **Photo**     | **CS**        | **Physics**   | **WikiCS**|
> |---------------|--------------|---------------|---------------|---------------|--------------|
> | **Polynormer**| 81,387,540   | 119,132,176 | 106,377,246 | 134,823,946 | 7,537,172 |
> | **G$^2$Former (Ours)** | 2,115,337 | 415,408 | 10,860,034 | 10,860,034 | 249,448 |
>
> **Table: Comparison of model parameter scales on heterophilous graphs.**
>
> | | **amazon\_ratings** | **roman\_empire** | **minesweeper** | **questions** |
> |----------------|---------------------|-------------------|-----------------|-----------------|
> | **Polynormer**| 9,112,586 | 156,053,540 | 167,983,108 | 105,619,460 |
> | **G$^2$Former (Ours)**| 2,777,122 | 5,958,692 | 213,141 | 8,029,700 |
>
> **W2.** We thank the reviewer for highlighting the need for a clearer conceptual description of our graph-guided filtering module. This undoubtedly strengthens the elaboration and demonstration of the method. The underlying intuition is as follows: The global-aware graph filter acts as a feature perturbation mechanism that introduces structured noise, guided by multi-channel spectral constraints. This“guided noise”is then injected into the node features via a residual-style update, similar in spirit to data augmentation or feature sharpening.
>
> It encourages the model to focus on features that are consistently highlighted across multiple frequency channels. This reduces over-reliance on spurious local signals and helps the model generalize better across different graph structures.
>
> We present the response of different spectral channels under varying levels of heterophily, where heterophily (**Hete.**) is defined as the proportion of heterophilous edges among all edges. The table below reports the $Dis(Dis_{hete}/Dis_{homo})$. $\mathbf{X}$ and $\mathbf{X'}$ represent the features before and after graph-guided filtering, respectively. $Dis_{hete}, Dis_{homo}$ denote the average Euclidean distances between node pairs connected by heterophilous and homophilous edges, $Dis=Dis_{hete}/(Dis_{hete}+Dis_{homo})$. $\mathcal{F}_i$ denotes the filter associated with the $i-$th channel, where a larger $i$ corresponds to higher-frequency bands. As shown in the table, for smaller graphs (Photo and Minesweeper), graph-guided filtering tends to enhance the distinction between heterophilous node pairs, resulting in increased $Dis$ scores in $\mathbf{X'}$ (feature sharpening). In contrast, for large-scale graphs (ogbn-products and pokec), the filtering operation exhibits a compressive effect on node attributes while preserving the original feature distribution, as evidenced by the simultaneous decrease in both $Dis\_{hete}$ and $Dis\_{homo}$ (data augmentation), with the overall $Dis$ score remaining largely unchanged. The model effectively adapts to both enhancement effects under varying levels of heterophily and graph scales. We sincerely thank you again for your meticulous and rigorous feedback.
>
> **Table:Response results of each channel on graphs with different heterophily.**
>
> | Dataset| **Photo** | **ogbn-products** | **pokec** | **Minesweeper** |
> |--------------|-----------|-------------------|-----------|-----------------|
> | Hete.| **0.1674** | **0.1887** | **0.5552** | **0.2815** |
> | **$\mathbf{X}$** | **0.5466 (0.07/0.06)**  | **0.5043 (37.80/37.16)** | **0.5299 (27.56/24.45)** | **0.5544 (0.97/0.78)** |
> | **$\mathcal{F}_0$** | **0.6835 (3.27/1.52)**  | **0.4895 (0.58/0.60)** | **0.5311 (1.02/0.90)** | **0.5855 (0.25/0.18)** |
> | **$\mathcal{F}_1$** | **0.6835 (6.55/3.03)**  | **0.4892 (1.72/1.80)** | **0.5310 (3.02/2.67)** | **0.5855 (0.96/0.68)** |
> | **$\mathcal{F}_2$**  | **0.6835 (3.27/1.52)**  | **0.4891 (1.72/1.80)** | **0.5310 (2.98/2.64)** | **0.5864 (1.40/0.98)** |
> | **$\mathcal{F}_3$**  | **-** | **0.4889 (0.57/0.60)**   | **0.5309 (0.98/0.87)**   | **0.5873 (0.90/0.63)** |
> | **$\mathcal{F}_4$**  | **-** | **-** | **-** | **0.5882 (0.22/0.15)** |
> | **$\mathbf{X'}$** | **0.6698 (12.36/6.09)** | **0.5000 (25.82/25.82)** | **0.5288 (14.69/13.09)** | **0.5835 (2.94/2.10)** |
>
> **W3.** We thank the reviewer for this valuable suggestion regarding section transitions and structural clarity. In the final version, we will add brief motivation paragraphs at the beginning of each major subsection.
> We believe these additions will significantly improve the readability and conceptual accessibility of the paper.
>
> **W4.** We appreciate the reviewer’s suggestion and understand the need for clearer framing. In fact, we would like to clarify that our categorization of Graph Transformer architectures is not introduced post-hoc, but rather follows the taxonomy established in Polynormer **[4]**. In other words, we followed the published work precisely to avoid unnecessary controversy.
> Specifically, as stated in lines 54–56 and illustrated in Figure 1, we describe SGFormer as a representative of the local-and-global paradigm, and Polynormer as a local-to-global model. We adopt this framing as a basis to highlight the novelty of our global-to-local design. Thank you again for your valuable suggestion.
>
> **W5.** We thank the reviewer for raising this important point.Regarding the selection of baselines, our primary goal was to benchmark G2Former against a wide range of state-of-the-art Graph Transformers and classical GNNs, rather than exhaustively cover all heterophily-specialized GNNs. In addition to H2GCN and FAGCN, we also provide experimental results (In the table below, our method consistently demonstrates superior performance across various benchmarks.) of CPGNN, GPRGNN, FSGNN, and GloGNN to address your concerns.
>
> We appreciate the reviewer’s observation, and we agree that G2Former does not achieve the best results on small citation networks such as Cora, Citeseer, and PubMed. This is not an oversight, but rather an expected and honest outcome — and reflects the underlying inductive bias of classical GNNs. The high homophily of these graphs aligns well with the low-pass filtering nature of GNNs. In the R1 of Reviewer ieX9, their homophily ranks among the top three under the homophilous benchmark. It is not surprising that G2Former — which incorporates high-frequency information — does not always outperform them on these tasks. However, this also reflects a broader issue in GNN benchmarking.  As observed in our experiments, many Graph Transformers perform well only on these small, homophilous datasets, but suffer from poor generalization on larger, more heterophilous graphs, which are in line with the conclusions of **[5]**. We hope this work provides useful insights and a reproducible benchmark for the community to better understand the true utility and limitations of Transformers in graph learning.
>
> **Table : Comparison of performance on heterophilous graphs. The best results are highlighted in bold.**
>
> |  | **Squirrel** | **Chameleon** | **Amazon-Ratings** | **Roman-Empire** | **Minesweeper** | **Questions** |
> |----------|--------------|---------------|---------------------|------------------|------------------|---------------|
> | Metric| Accuracy | Accuracy | Accuracy | Accuracy | ROC-AUC | ROC-AUC |
> | **H2GCN** | 34.75${\pm}$0.9 | 27.80${\pm}$3.8 | 36.40${\pm}$0.3 | 60.01${\pm}$0.4 | 88.65${\pm}$0.3 | 62.37${\pm}$1.3 |
> | **FAGCN** | 36.33${\pm}$1.6 | 34.67${\pm}$3.9 | 40.28${\pm}$0.6 | 62.89${\pm}$0.7 | 50.19${\pm}$1.3 | 67.99${\pm}$1.5 |
> | **CPGNN** | 30.01${\pm}$2.1 | 32.75${\pm}$3.7 | 38.66${\pm}$0.6 | 62.37${\pm}$0.4 | 52.01${\pm}$0.7 | 65.97${\pm}$1.6 |
> | **GPRGNN**| 39.01${\pm}$2.6 | 39.64${\pm}$3.8 | 45.02${\pm}$0.3 | 65.13${\pm}$0.7 | 85.26${\pm}$0.5 | 54.38${\pm}$1.4 |
> | **FSGNN** | 34.99${\pm}$1.2 | 40.33${\pm}$3.5 | 51.87${\pm}$0.8 | 79.65${\pm}$0.4 | 90.11${\pm}$0.7 | 78.85${\pm}$0.9 |
> | **GloGNN** | 35.36${\pm}$2.1 | 23.99${\pm}$4.3 | 36.97${\pm}$0.1 | 60.21${\pm}$0.6 | 51.26${\pm}$1.2 | 65.32${\pm}$1.2 |
> | **Ours**  | **45.53**${\pm}$1.7 | **44.34**${\pm}$4.3 | **55.86**${\pm}$0.3 | **93.03**${\pm}$0.3 | **99.45**${\pm}$0.2 | **79.55**${\pm}$0.1 |
>
> [1]  Attention is all you need. NeurIPS 2017.
>
> [2] Adaptive universal generalized pagerank graph neural network. ICLR 2020.
>
> [3] Rethinking graph neural networks for anomaly detection. ICML 2022.
>
> [4] Polynormer: Polynomial-Expressive Graph Transformer in Linear Time The Twelfth International Conference on Learning Representations. ICLR 2024.
>
> [5] Position: Graph Learning Will Lose Relevance Due To Poor Benchmarks. ICML 2025.

---

> > ### Comment · Reviewer_3Sa7 · 2025-08-05
> >
> > Thank you for your detailed responses to my review. I appreciate the clarifications and additional results you provided. They addressed several of my questions and helped me better understand key aspects, such as the use of two FFNs, the paradigm framing, and the inclusion of additional heterophily-focused GNN baselines.
> >
> > I do have a question regarding the reported performance on the Squirrel and Chameleon datasets. These datasets are known to contain many duplicate or nearly identical nodes, as highlighted in the paper “A Critical Look at the Evaluation of GNNs under Heterophily: Are We Really Making Progress?” (ICLR 2023), which argues that results on these versions may be biased and not reflective of true generalization performance.
> >
> > Could you clarify:
> > - Which version of Squirrel and Chameleon was used in your experiments?
> > - Have you evaluated G2Former on the **cleaned** versions of these datasets?
> > - If not, would you consider running this evaluation or commenting on how you expect G2Former to behave under the cleaned setup?
> >
> > This would help contextualize your strong results on these benchmarks and further strengthen your conclusions regarding performance under heterophily.
> >
> > Thank you again for your engaging and thoughtful response.

---

> > > ### Author Response · Authors · 2025-08-06
> > >
> > > Thank you for your thoughtful follow-up and for raising this important point regarding the evaluation on Squirrel and Chameleon.
> > >
> > > We would like to clarify that in our submission, we already use the **cleaned versions of both datasets** — i.e., the versions proposed in [41], which remove duplicate or near-duplicate nodes. This is stated in our submission (Line 241-242 in Section 4):
> > >
> > > “Note that for heterophilous datasets, we use the new version [41] of Chameleon and Squirrel, which removes overlapped nodes.”
> > >
> > > We agree that evaluating on these cleaned datasets is important for ensuring fair and meaningful benchmarking under heterophily, and we appreciate your attention to this detail.
> > >
> > > Thank you again for your kind words and for engaging deeply with our work.
> > >
> > > [41] A Critical Look at the Evaluation of GNNs under Heterophily: Are We Really Making Progress? ICLR 2023.

---

> > > > ### Comment · Reviewer_3Sa7 · 2025-08-08
> > > >
> > > > Thank you for the clarification regarding the use of the cleaned versions of Squirrel and Chameleon. I appreciate your pointing out the reference in Line 241–242 and confirming that the evaluation was conducted on the de-duplicated datasets. This helps reinforce the credibility of your results on those benchmarks.
> > > >
> > > > After considering all aspects of the paper, including the paper's clarity, method's adaptability across homophily, I’ve decided to keep my score unchanged. Thank you again for your thoughtful and thorough responses addressing my questions.

---

> ### Author Response · Authors · 2025-08-08
>
> Thank you very much for your careful assessment and for acknowledging the clarity we provided regarding dataset preparation and evaluation. This carries important implications for **answering the empirically contested question of the actual role of Transformers in GNNs, as well as for highlighting the potential of classical GNNs on homophilous graphs**. We will refine the presentation to avoid any potential misunderstandings. We sincerely appreciate your thoughtful feedback and engagement with our work.

---

> ### Author Response · Authors · 2025-08-08
>
> Thank you for your thoughtful feedback. We sincerely appreciate your comments regarding both the clarity of the paper and the adaptability of our method on homophilous graphs.
>
> **a.** Regarding clarity, We will revise the content to ensure that each part clearly links the observed challenges to our design choices, as you suggested.
>
> **b.** In evaluating on homophilous datasets such as Cora, Citeseer, and PubMed, **we wish to emphasize that the slightly lower scores of our method on these datasets are expected, and we hope this observation will draw the community’s attention.**
>
> The following presents the edge discriminability under homophilous settings, computed as $Dis=Dis_{hete}/(Dis_{hete}+Dis_{homo})$. $Dis_{hete}, Dis_{homo}$ denote the average Euclidean distances between node pairs connected by heterophilous and homophilous edges, respectively.  In the presence of highly homophilous graphs—such as Cora, Citeseer, and PubMed, which rank among the **top three** in terms of homophily—classical GNNs are often sufficient (Table 1 in Section 4), as their inherent low-pass filtering behavior naturally aligns with the underlying graph structure. **This observation empirically supports our central claim that attention mechanisms struggle to help GNNs surpass their inherent performance limits.**
>
> **Table: Edge discrimination ($Dis$) on homophilous graphs.**
>
> | Dataset| Cora    | Citeseer | PubMed  | Computer | Photo   | cs      | Physics | WikiCS  |
> |-----------|---------|----------|---------|----------|---------|---------|---------|---------|
> | $Dis$    | **0.5749** | **0.6060** | **0.5742** | 0.4811   | 0.4501  | 0.5156  | 0.5154  | 0.5281  |
>
> Additionally, as discussed in the paper (Line 258-260 in Section 4.1), these legacy benchmarks have been widely overused and heavily tuned against, often leading to overfitting. For example, while **SGFormer** achieves the best performance on these three datasets, it underperforms on the remaining homophilous, heterophilous, and large-scale graphs. In contrast, **Polynormer** explicitly avoids these three benchmarks in its original paper. **We did not specifically optimize for these datasets, but rather investigated the actual role of attention mechanisms in GNNs through a comprehensive evaluation across 17 homophilous, heterophilous, and large-scale graphs.** In addition, beyond these three benchmarks, our method’s performance on the remaining 7 homophilous graphs—especially on **ogbn-products (a homophilous graph with over two million nodes and 47 classes, Table 3 in Section 4.1)**—also demonstrates its adaptability to homophily.
>
> We respectfully hope that our explanation has clarified our perspective. Thank you once again for your thoughtful attention to our work.

---

### Official Review · Reviewer_ieX9 · 2025-07-03

**Clarity:** 4
**Significance:** 3
**Originality:** 3
**Rating:** 5
**Confidence:** 4

**Summary:**

This paper presents G2Former, a graph transformer architecture that integrates global self-attention with local GNN in a **global-to-local** manner. The key idea is to treat the transformer as a band-pass graph filter that injects guided noise into node features across multiple frequency channels, which a downstream GNN then aggregates. The authors hypothesize that while pure transformers may not exceed the inherent performance of GNNs, a restricted multi-channel transformer can improve GNN generalization by capturing both low and high-frequency graph signals that classic GNNs (mostly low-pass) might miss. They support this claim with theoretical analysis and extensive node classification experiments on diverse graph datasets.

**Questions:**

Had fun reading the paper! Thanks to the authors for their clear language and apt amount of rigor. I have a few questions that I think would help clarify the contributions and applicability of G2Former.
1. Performance vs simplicity: Given that G2Former's accuracy improvements over well-tuned GNNs are quite small (often < 0.5%), can the authors elaborate on when using G2Former is most justified? For what types of graphs or scenarios would the added complexity pay off most? It would help to know if there are specific regimes (e.g., strong heterophily, extremely large and sparse graphs, etc.) where G2Former yields a clear advantage, as opposed to cases where a simple GNN is "good enough".
2. Channel initialization and sensitivity: The paper mentions two channel initialization strategies and provides a sensitivity analysis on the number of channels in Appendix B.1. Could the authors clarify the following: (a) what are these two strategies in intuitive terms? and which one was ultimately used for the main results? (b) How sensitive is G2Former to the choice of the total number of channels $(\alpha + \beta)$ or the distribution of those channels between low vs high frequency? For instance, if one uses too few channels or suboptimal $(\alpha,\beta)$ combinations, does performance degrade notably?
3. Understanding "guided noise": The concept of guided noise is intriguing. Could the authors provide more intuition or examples of what the transformer-generated noise looks like in practice? For example, do the attention-based features highlight anomalous or hard-to-predict nodes, or do they act as a form of dropout/regularization on the feature space? Any qualitative or quantitative insight into what the multi-channel attention is learning (perhaps via attention weights or frequency response curves) would help justify why this "noise" improves generalization.
4. Limits of the transformer component: The authors conclude that G2Former "falls short of surpassing the performance ceiling of GNNs". What do the authors see as the fundamental limitation that prevents the transformer from exceeding GNN performance? Is it because the transformer's global attention, even when band-limited, cannot add new information beyond what multi-hop message passing provides? Or could it be an optimization issue (e.g., harder to train the attention fully)? Understanding this would be valuable for future research – for instance, if the issue is optimization, maybe different training schemes could help, whereas if it's an inherent limitation of global filters, then the community might better focus elsewhere.
5. Attention sparsity and interpretability: G2Former currently constructs a dense attention graph (complete graph) but leverages a low-rank factorization to apply it efficiently. Do the authors have a sense of whether the learned attention matrix is sparse or diffuse? For example, do nodes attend strongly only to a limited set of other nodes (implying a sparser effective structure), or is the attention more spread out? Any information on this would help interpret how the model is functioning. Additionally, can any of the learned global connections be interpreted as meaningful (e.g., connecting nodes of the same class across clusters, which a local GNN might not connect)? Providing a bit of interpretation could strengthen the argument that the global filter is doing something non-trivial and useful (again, not expecting too much work here, please feel free to tell me if this is too much work and is out of scope of the current project, it will not affect my rating/view of this work).

**Ethical Concerns:**

["NO or VERY MINOR ethics concerns only"]

**Limitations:**

Limitations have been addressed adequately.

**Paper Formatting Concerns:**

No concerns.

**Quality:**

3

**Strengths And Weaknesses:**

Strengths
- Global-to-local design: G2Former integrates a global transformer as a band-pass filter feeding into a GNN, which avoids redundancy and leverages complementary spectral information.
- Theoretical grounding: The model is formally shown to act as a multi-channel polynomial filter over a learned Laplacian, with provable ability to span arbitrary frequency bands via $(\alpha, \beta)$ (Theorem 3.1).
- Scalability: The architecture maintains linear complexity in $|V| + |E|$ using low-rank attention and polynomial filtering, and runs efficiently on large-scale graphs like ogbn-products etc.
- Strong experiments: Evaluated on 17 diverse graphs, the model shows consistent improvements over both classical GNNs and graph transformers, with well-tuned baselines and rigorous ablations.
- Regularization effect: The multi-channel setup stabilizes learning and improves test generalization, as shown via controlled attention-only ablations and loss curves.

Weaknesses
- Marginal gains on homophilic graphs: Improvements over strong GNN baselines are often under 0.5%, and in some cases, negligible or absent. It would be interesting to address why the method is not more effective on these kinds of datasets.
- Added complexity: The multi-channel design, spectral filtering, and channel hyperparameters introduce non-trivial implementation and tuning overhead for modest gains. Is there a simpler alternative that could achieve similar results? I would expect a simpler model to mimic these effects without the spectral complexity.
- Guided noise concept lacks clarity: The role and behavior of transformer-injected "noise" are not fully explained or visualized, which makes interpretation more challenging.
- Limited task diversity: Evaluation is restricted to node classification; it's unclear if the method generalizes to link prediction or graph-level tasks. The nature of the global-to-local design could be more thoroughly explored across different graph learning tasks. Of course, this would be a separate paper, but it would be interesting to comment on it a little if you have some thoughts. (Not expecting any analysis or results here, so feel free to ignore this comment.)

---

> ### Author Rebuttal · Authors · 2025-07-30
>
> **Q1**. Thank you for this thoughtful question. We agree that G2Former does not always outperform well-tuned GNNs by large margins, especially on simple or homophilous graphs. This observation aligns with our central motivation: To examine whether and when global attention mechanisms truly help GNNs generalize — not just whether they boost accuracy in absolute terms. Our experiments (Table 1-3, Figure 4) show that G2Former tends to offer the most tangible benefits in the following regimes:Heterophilous graphs, where spatial domain neighborhood are noisy or conflicting; Large and sparsely connected graphs, where global signal propagation improves convergence and stability. G2Former offers a principled framework for interpreting and refining Graph Transformers through spectral constraints. The following presents the link discriminability under homophilous settings, computed as $Dis=Dis_{hete}/(Dis_{hete}+Dis_{homo})$. $Dis_{hete}, Dis_{homo}$ denote the average Euclidean distances between node pairs connected by heterophilous and homophilous edges, respectively. The dataset with the smallest performance improvement exhibits the highest $Dis$ score, indicating lower similarity among node pairs connected by heterophilous edges, and consequently, a higher degree of homophily in the graph structure. This is in line with your judgment.
>
> **Table: Link discrimination ($Dis$) on homophilous graphs.**
>
> | Dataset   | Cora    | Citeseer | PubMed  | Computer | Photo   | cs      | Physics | WikiCS  |
> |-----------|---------|----------|---------|----------|---------|---------|---------|---------|
> | $Dis$    | **0.5749** | **0.6060** | **0.5742** | 0.4811   | 0.4501  | 0.5156  | 0.5154  | 0.5281  |
>
>
> **Q2**. Thank you for raising this important and thoughtful question. We are happy to clarify the two initialization strategies and the robustness of channel design. (a) Initialization strategies: As described in Figure 3, we consider two initialization strategies for the learnable spectral filters: Full-spectrum initialization: channels are initialized to evenly cover the full frequency domain (from low to high); Low-spectrum initialization: all channels are initialized within the low-frequency range. The full-spectrum initialization is used in all main experiments (Tables 1-3), as it consistently outperforms the low-spectrum alternative across both homophilous and heterophilous graphs (Figure 6).
> (b) Sensitivity to the total number of channels $T = \alpha + \beta$: As shown in Figure 7 and discussed in Appendix B.1, G$^2$Former exhibits dataset-specific preferences for $T$, but overall demonstrates robustness to its precise value: On homophilous graphs, larger $T$ (e.g., $T = 7$ on WikiCS) help suppress spurious attention due to oversmoothing, effectively regulating the attention spectrum; On heterophilous graphs, smaller $T$ (e.g., $T = 1$ on Amazon-Ratings) are preferred, avoiding over-constraining the attention module, which could otherwise limit expressiveness. We also show in Figures 8 and 9 that larger $T$ values may introduce redundant channels and optimization challenges, but even suboptimal choices of $T$ still yield test performance comparable to or better than strong GNN baselines.
>
> **Q3**. We thank the reviewer for their insightful interest in the concept of guided noise, which indeed plays a central role in our interpretation of G$^2$Former's generalization behavior. At a high level, we use “guided noise” to refer to structured spectral perturbations introduced by multi-channel attention filters — which differ from random noise in that they are constrained to lie within specific frequency bands (e.g., low-pass, high-pass). We present the response of different spectral channels under varying levels of heterophily, where heterophily (**Hete.**) is defined as the proportion of heterophilous edges among all edges. The table reports the $Dis(Dis_{hete}/Dis_{homo})$, for $\mathbf{X}$ and $\mathbf{X'}$, representing the features before and after graph-guided filtering, respectively. $\mathcal{F}_i$ denotes the filter associated with the $i-$th channel, where a larger $i$ corresponds to higher-frequency bands. As shown in the table, for smaller-scale graphs such as Photo and Minesweeper, graph-guided filtering tends to enhance the distinction between heterophilous node pairs, resulting in increased $Dis$ scores in $\mathbf{X'}$ . In contrast, for large-scale graphs such as ogbn-products and pokec, the filtering operation exhibits a compressive effect on node attributes, reducing both $Dis\_{hete}$ and $Dis\_{homo}$.
> Nevertheless, the overall feature distribution remains close to the original, as reflected by similar $Dis$ scores before and after filtering. These findings highlight the adaptive role of graph-guided filtering: enhancing local discriminability in smaller graphs, while promoting feature compactness in larger graphs without significantly distorting the global structure.
>
> **Table:Response results of each channel on graphs with different heterophily.**
>
> | **Dataset**  | **Photo** | **ogbn-products** | **pokec** | **Minesweeper** |
> |--------------|-----------|-------------------|-----------|-----------------|
> | **Hete.**    | **0.1674** | **0.1887** | **0.5552** | **0.2815** |
> | **$\mathbf{X}$**         | **0.5466 (0.07/0.06)**  | **0.5043 (37.80/37.16)** | **0.5299 (27.56/24.45)** | **0.5544 (0.97/0.78)** |
> | **$\mathcal{F}_0$**      | **0.6835 (3.27/1.52)**  | **0.4895 (0.58/0.60)**   | **0.5311 (1.02/0.90)**   | **0.5855 (0.25/0.18)** |
> | **$\mathcal{F}_1$**      | **0.6835 (6.55/3.03)**  | **0.4892 (1.72/1.80)**   | **0.5310 (3.02/2.67)**   | **0.5855 (0.96/0.68)** |
> | **$\mathcal{F}_2$**      | **0.6835 (3.27/1.52)**  | **0.4891 (1.72/1.80)**   | **0.5310 (2.98/2.64)**   | **0.5864 (1.40/0.98)** |
> | **$\mathcal{F}_3$**      | **-**                  | **0.4889 (0.57/0.60)**   | **0.5309 (0.98/0.87)**   | **0.5873 (0.90/0.63)** |
> | **$\mathcal{F}_4$**      | **-**                  | **-**                    | **-**                    | **0.5882 (0.22/0.15)** |
> | **$\mathbf{X'}$**        | **0.6698 (12.36/6.09)** | **0.5000 (25.82/25.82)** | **0.5288 (14.69/13.09)** | **0.5835 (2.94/2.10)** |
>
> **Q4.** We deeply appreciate this question, as it aligns closely with the central motivation of our study. Based on extensive empirical exploration, we believe that the inability of global attention mechanisms to surpass well-tuned GNNs stems primarily from their inherent limitations in modeling graph inductive bias, rather than optimization difficulties. Specifically, our findings suggest that attention-based modules struggle to capture the complex, often irregular topological priors inherent in real-world graphs. Unlike Euclidean domains (e.g., images or text), graph connectivity does not necessarily follow intuitive patterns like “similar nodes are more likely to be connected.” Moreover, a recent empirical study **[1]** also shows that graph Transformers exhibit no significant advantage over classical GNNs on graph-level tasks, further reinforcing our conclusion. This raises fundamental concerns about the assumption that global attention can automatically discover useful relational structure. Our results suggest that global attention, even when spectrally constrained, cannot reliably introduce valuable new connections beyond what multi-hop GNNs already aggregate. Therefore, we share the reviewer’s view that this may reflect a fundamental limitation of current global filtering approaches. Rather than solely developing more complex architectures, we advocate for a shift in focus back to the analysis and modeling of graph topology itself, which may offer more fruitful directions for advancing the field.
>
> **Q5.** We appreciate the reviewer’s thoughtful comments regarding the sparsity and interpretability of the attention matrix. As in prior works such as SGFormer and Polynormer, G$^2$Former employs a dense attention mechanism. This design ensures scalability and fairness in comparisons, especially on large graphs where explicit sparsification would incur high computational cost. While the learned attention matrix is mathematically dense, we observe that the effective attention tends to be soft-sparse, i.e., each node often focuses more strongly on a subset of other nodes (Figure 5).  Regarding interpretability, we do not rule out the possibility that G$^2$Former captures semantically meaningful global connections, such as linking same-class nodes across clusters. However, our primary conclusion is based on entire performance trends: these potential benefits are often offset by the topological noise introduced by dense global attention, especially on homophilous graphs. Moreover, such connections may be  due to the fact that the node pair is a training sample, and their generalization to unseen test pairs remains uncertain. While a detailed interpretability analysis is beyond the current paper’s scope, we agree that exploring this further would be a promising direction for future work. In ongoing work, we are exploring relaxed message-passing schemes that aim to combine the strengths of local GNN modeling and global perception, seeking to expand the receptive field while mitigating the topological noise introduced by dense attention. Thank you very much for your support.
>
> [1] Can Classic GNNs Be Strong Baselines for Graph-level Tasks? Simple Architectures Meet Excellence. ICML 2025.

---

### Note · Authors · 2025-08-12

We thank ACs and the reviewers for their time, effort, and constructive feedback. The discussion phase has been very helpful in clarifying our contributions and improving the presentation of the paper.

This work is driven by a central research question: Can attention mechanisms truly help GNNs overcome their performance bottlenecks? This issue has been challenged by an empirical study [1]. Our investigation yields a nuanced conclusion: Empirically, relying on attention mechanism often fails to help GNNs break through their performance bottlenecks; however, with appropriate spectral constraint, it can enhance generalization of GNNs—particularly on highly heterophilous, and large-scale graphs. Our findings provide empirical evidence and methodological insight into the role of attention in GNNs.

Our method achieves further generalization improvements on most benchmarks—including performance metrics, test loss curves, visualization of the attention matrix under the constraint, and generalization gaps—and we deliberately present results on the three overused high-homophily datasets: Cora, Citeseer, and PubMed. While some prior works have reported state-of-the-art results on these benchmarks, they underperform on others, underscoring the potential evaluation risk of tuning to a small set of benchmarks (as noted in Section 4.1, Lines 258–260 of our paper). In contrast, our emphasis is on achieving robust cross-dataset generalization, which allows us to demonstrate more pronounced advantages on modern, diverse, and more challenging benchmarks, such as million-node graphs.

We appreciate the positive feedback on clarity and agree with the suggestion to begin each technical section with a brief motivation paragraph. We will revise the manuscript to strengthen transitions between sections, effectively linking the observed limitations to our design choices—for example, the transition from classical spectral graph filters to attention-based global-perception graph filters, as well as the introduction of multi-channel band-pass filtering constraints to reduce topological noise interference introduced by attention.

We once again sincerely thank ACs and the reviewers for their constructive comments and close attention.

[1] Classic GNNs are strong baselines: Reassessing GNNs for node classification. NeurIPS 2024.

---

### Decision · Program_Chairs · 2025-09-17

**Decision:**

Accept (poster)

**Comment:**

This paper investigates the fundamental question of whether Transformers can help Graph Neural Networks overcome their performance bottlenecks by proposing G2Former, which reinterprets global attention as band-pass graph filters with multi-channel spectral constraints. The work addresses a timely and important challenge in the graph learning community, motivated by recent empirical evidence questioning the effectiveness of Graph Transformers. The four reviewers provided scores of 5, 3, 4, and 5, with generally positive feedback on the experimental comprehensiveness but concerns about clarity and theoretical depth. Key strengths include: (1) Comprehensive empirical investigation - The evaluation spans 17 diverse datasets across homophilous, heterophilous, and large-scale graphs, providing robust evidence for the proposed approach's broad applicability; (2) Novel theoretical perspective - The reinterpretation of attention mechanisms as learnable spectral filters with band-pass characteristics offers a principled framework for understanding and constraining global attention in GNNs; (3) Honest and nuanced findings - The authors candidly report that G2Former does not always outperform baselines, particularly on highly homophilous datasets, while demonstrating clear benefits on heterophilous and large-scale graphs; (4) Important research question - The work tackles the fundamental question of whether attention mechanisms truly enhance GNN performance, providing empirical evidence that challenges prevailing assumptions in the field. Limitations acknowledged: (1) Modest performance improvements - The gains are often marginal (< 0.5%), though the authors argue their focus is on understanding the role of attention rather than achieving state-of-the-art results; (2) Presentation challenges - Multiple reviewers noted clarity issues and dense mathematical exposition, though the authors committed to improving section transitions and motivation; (3) Limited theoretical depth - While the spectral interpretation is novel, the theoretical understanding of generalization mechanisms remains primarily empirical. The authors demonstrated strong engagement during the review process, addressing technical concerns and providing additional experiments including comparisons with heterophily-focused baselines and generalization gap analyses. The work makes a valuable contribution to the graph learning community by providing a principled framework for understanding attention mechanisms and offering insights into when and why global attention may benefit GNN generalization, particularly under structural noise or heterophily.